# PKA regulatory subunit Bcy1 couples growth, lipid metabolism, and fermentation during anaerobic xylose growth in *Saccharomyces cerevisiae*

Ellen R. Wagner[ORCID][1,2,3], Nicole M. Nightingale[2,4], Annie Jen[2,4], Katherine A. Overmyer[4,5,6], Mick McGee[2], Joshua J. Coon[2,3,4,5,6,7], Audrey P. Gasch[ORCID][1,2,3]*

1 Laboratory of Genetics, University of Wisconsin-Madison, Madison, Wisconsin, United States of America, 2 Great Lakes Bioenergy Research Center, University of Wisconsin-Madison, Madison, Wisconsin, United States of America, 3 Center for Genomic Science Innovation, University of Wisconsin-Madison, Madison, Wisconsin, United States of America, 4 Department of Biomolecular Chemistry, University of Wisconsin-Madison, Madison, Wisconsin, United States of America, 5 Morgridge Institute for Research, Madison, Wisconsin, United States of America, 6 National Center for Quantitative Biology of Complex Systems, Madison, Wisconsin, United States of America, 7 Department of Chemistry, University of Wisconsin-Madison, Madison, Wisconsin, United States of America

* agasch@wisc.edu

**Data Availability Statement:** Raw and processed RNAseq files have been submitted to the NIH GEO database under project number GSE220465. Raw

## Abstract

Organisms have evolved elaborate physiological pathways that regulate growth, proliferation, metabolism, and stress response. These pathways must be properly coordinated to elicit the appropriate response to an ever-changing environment. While individual pathways have been well studied in a variety of model systems, there remains much to uncover about how pathways are integrated to produce systemic changes in a cell, especially in dynamic conditions. We previously showed that deletion of Protein Kinase A (PKA) regulatory subunit *BCY1* can decouple growth and metabolism in *Saccharomyces cerevisiae* engineered for anaerobic xylose fermentation, allowing for robust fermentation in the absence of division. This provides an opportunity to understand how PKA signaling normally coordinates these processes. Here, we integrated transcriptomic, lipidomic, and phospho-proteomic responses upon a glucose to xylose shift across a series of strains with different genetic mutations promoting either coupled or decoupled xylose-dependent growth and metabolism. Together, results suggested that defects in lipid homeostasis limit growth in the *bcy1Δ* strain despite robust metabolism. To further understand this mechanism, we performed adaptive laboratory evolutions to re-evolve coupled growth and metabolism in the *bcy1Δ* parental strain. The evolved strain harbored mutations in PKA subunit *TPK1* and lipid regulator *OPI1*, among other genes, and evolved changes in lipid profiles and gene expression. Deletion of the evolved *opi1* gene partially reverted the strain's phenotype to the *bcy1Δ* parent, with reduced growth and robust xylose fermentation. We suggest several models for how cells coordinate growth, metabolism, and other responses in budding yeast and how restructuring these processes enables anaerobic xylose utilization.

and processed lipidomic files were deposited in MassIVE database under dataset number MSV000090868.

**Funding:** This material is based upon work supported by the Great Lakes Bioenergy Research Center, U.S. Department of Energy, Office of Science, Office of Biological and Environmental Research under Award Number DE-SC001840, funding APG. ERW is supported by the National Science Foundation Graduate Research Fellowship Program under Grant No. DGE-1747503. Any opinions, findings, and conclusions or recommendations expressed in this material are those of the author(s) and do not necessarily reflect the views of the National Science Foundation. ERW was also supported by the Graduate School and the Office of the Vice Chancellor for Research and Graduate Education at the University of Wisconsin-Madison with general funding from the Wisconsin Alumni Research Foundation. The funders had no role in study design, data collection and analysis, decision to publish, or preparation of the manuscript.

**Competing interests:** JJC is a consultant for Thermo Fisher Scientific.

## Author summary

All organisms utilize an energy source to generate the cellular resources needed to grow and divide. Individual processes have been well studied, but the coordination and cross-talk between the process is not well understood. To study growth and metabolism coupling, we used a yeast strain that was genetically engineered to ferment the sugar xylose but lacked growth on the sugar. The decoupled growth and metabolism was caused by a deletion of a single gene in a highly conserved signaling pathway found in all eukaryotes. While our work is focused on xylose metabolism, we address the fundamental question of how cells coordinate growth with metabolism under non-ideal conditions. We identified changes in gene expression that implicated altered regulatory mechanisms involved in lipid metabolism correlating with decoupled growth and metabolism. Our work highlights the complexity of engineering new cellular functions and that global regulatory modifications, rather than altering individual pathways, may be required for broad cellular changes.

## Introduction

Many physiological processes are essential for growth, but so too is the coordination of those processes to form an integrated cellular system. Actively dividing cells must coordinate metabolism and division with the synthesis and segregation of DNA, proteins, organelles, and other macromolecules, all within a precisely timed cell cycle. Failure to coordinate these processes can jeopardize fitness due to suboptimal cellular composition and energy expenditures. Mechanistically, much remains unknown about how cells coordinate cellular processes. One of the best studied examples is the intimate control of successive cell cycle phases, which depends on interconnected transcriptional and post-translational controls regulated by dispersed checkpoints along the way [1–8]. The cell cycle is also coordinated with metabolism; cell-cycle regulators coordinate metabolic flux with cell-cycle phases, which may be related to cell size checkpoints since cells must reach a critical size before a new cell cycle is initiated [2,3,6–8]. A critical feature of integrated cellular systems is thus balancing energy demands with division and replication.

Knowing how cells coordinate growth and division with other physiological processes is important for understanding how cells function on a fundamental level, but it also has practical applications. Microbes can be engineered to produce a variety of commodity chemicals and biofuels with high yields to maximize economic returns. Microbial design strategies have considered how cells allocate resources so as to redirect cellular energy toward making compounds of interest [9–15]. Redirecting resources away from other processes can improve cellular product yields and thus decrease costs [16,17]. However, an added complication is that many industrial processes are stressful for engineered microbes, which mount stress-defense systems that further deplete cellular resources from product formation. The interplay of growth, metabolism, division, and stress defense remain murky, limiting engineering efforts [18,19].

Here, we studied how growth, division, and metabolism are normally coupled in cells by investigating a strain in which these processes have been decoupled. We previously characterized a series of *Saccharomyces cerevisiae* strains engineered to produce biofuel products from xylose, a pentose sugar abundant in plant biomass but not recognized by *S. cerevisiae* as a fermentable sugar [17,20–22]. A major goal for sustainable biofuel production is to utilize xylose

and other carbon sources to maximize biomass conversion to products. Past work in our center found that engineering *S. cerevisiae* to ferment xylose anaerobically requires core xylose metabolism genes (encoding xylose isomerase, xylulokinase, and transaldolase [23,24]); however, introducing these genes is not enough to enable fermentation. Many groups have combined strain engineering with adaptive laboratory evolution to evolve xylose fermentation [25–30]. Cells require additional null mutations in oxidoreductase *GRE3*, iron-sulfur (Fe-S) chaperone *ISU1*, and RAS signaling inhibitor *IRA2* [27,31]. We previously showed that these mutations help to rewire cellular signaling to unnaturally upregulate the growth-promoting Protein Kinase A (PKA) pathway in conjunction with Snf1 that usually responds to poor carbon sources [32]. We proposed that activating PKA and Snf1 promotes growth in the context of an otherwise unrecognized carbon source [32]. Coordinated induction of PKA and Snf1 allows cells to recognize xylose as a fermentable carbon source while enhancing growth and metabolism signals.

Although PKA activation is critical for anaerobic xylose growth and metabolism, during that study we made a surprising discovery: the mechanism of PKA up-regulation influences how growth and metabolism are coordinated. PKA can be activated by RAS activity, which stimulates adenylate cyclase to produce the allosteric regulator cAMP that binds and dissociates the PKA regulatory subunit Bcy1 (Fig 1A) [33]. In engineered yeast, activating PKA by *IRA2* deletion, thus increasing RAS activity, enables rapid anaerobic xylose fermentation and growth on xylose as the sole carbon source. However, activating PKA by deleting the PKA regulatory subunit *BCY1* allows rapid anaerobic xylose fermentation but with little to no growth (Fig 1B and 1C) [32]. In both strains, the effect is due to PKA upregulation since inhibition of PKA activity blocked both metabolism and growth [32]. Thus, deleting *BCY1* in this strain background decouples growth and xylose metabolism for reasons that are not known. Importantly, other uncoupled biological processes related to PKA function have also been described [34], implying PKA's central role in process coupling.

Here, we explored phenotypic consequences of *IRA2* and *BCY1* deletions to elucidate how cells normally coordinate growth and metabolism. We integrated transcriptomic, phospho-proteomic, and lipidomic analysis across a suite of strains with different mutations and growth/metabolism phenotypes. The results implicated the importance of lipid metabolism as a linchpin in the coordination of growth with metabolism: cells lacking *BCY1* show unique transcriptomic and lipidomic responses that point to defects in lipid regulation. To uncover causal genes, we also performed adaptive evolution to re-evolve growth coordination in the *bcy1Δ* strain. Remarkably, the evolved strain acquired mutations in a PKA catalytic subunit *TPK1* and phospholipid biosynthesis regulator *OPI1*, among other genes, and Opi1 was required for the growth-metabolism coupling in the evolved strain. These results suggest that PKA-dependent regulation of lipid metabolism is critical for growth, perhaps to coordinate membrane biogenesis and signaling with other cellular processes.

## Results

We began by characterizing a suite of strains with different anaerobic xylose growth and fermentation capabilities. Parental strain Y184 harbors the xylose-metabolism gene cassette along with mutations in *ISU1* and *GRE3* but cannot grow on or metabolize xylose anaerobically (Fig 1B and 1C). Deleting *IRA2* from this strain allows cells to grow on and metabolize xylose anaerobically. In contrast, deletion of *BCY1* from Y184 permits rapid anaerobic xylose fermentation but with only minimal growth (Fig 1B, 1C, and 1E). Previous work studying growth over 90 hours validated limited growth of the *bcy1Δ* strain even after long periods [35]. We also investigated an *ira2Δbcy1Δ* double mutant. The double mutant was phenotypically similar

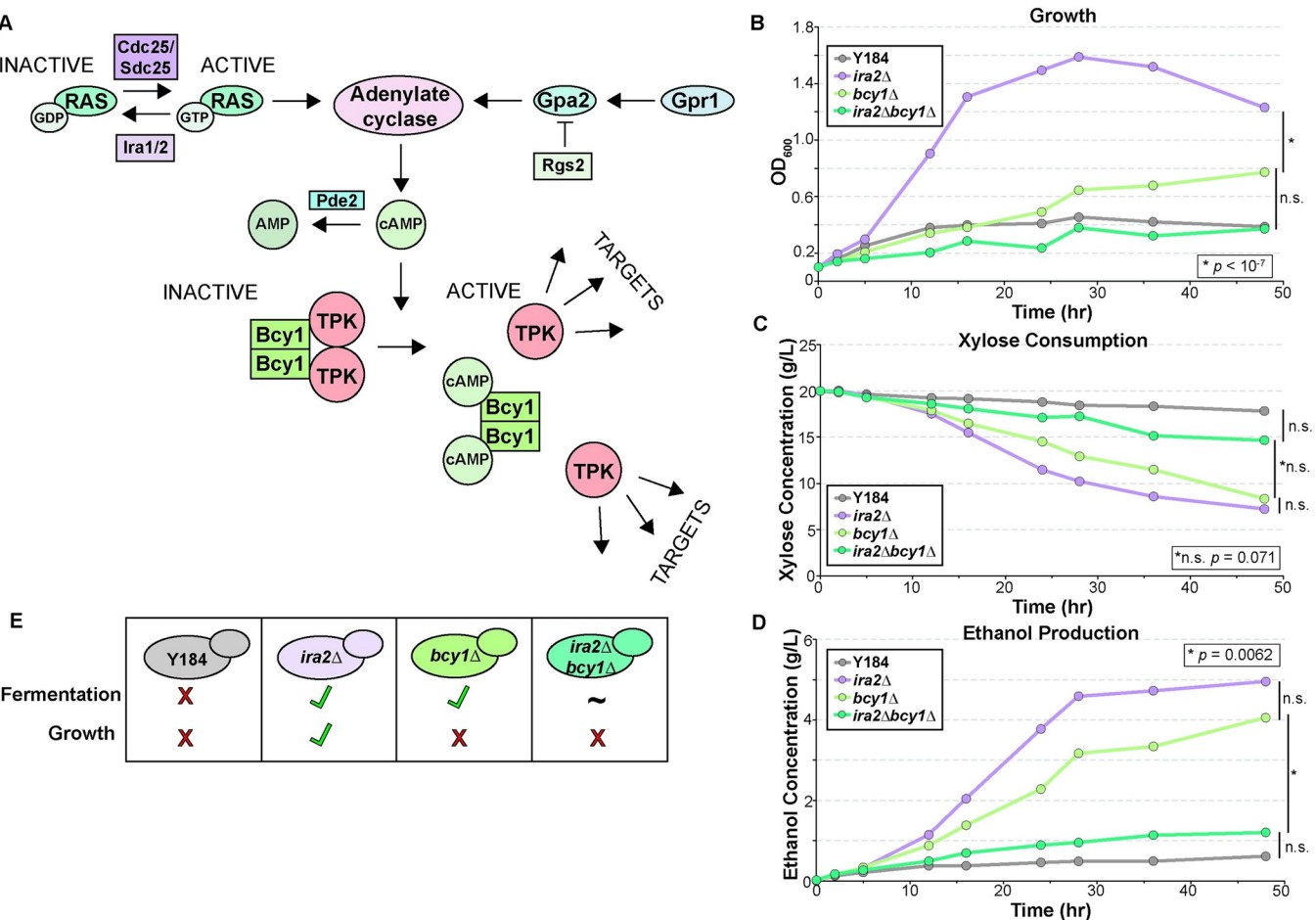

**Fig 1. Activation of PKA is needed for xylose fermentation. A**. A brief overview of the PKA signaling pathway. **B-C.** Average (n = 6 biological replicates) growth (OD600, optical density) and (**B**) xylose concentration in the medium over time of parental Y184, *ira2Δ*, *bcy1Δ*, and *ira2Δbcy1Δ* strains grown anaerobically on rich medium containing xylose as a carbon source. Asterisks denote significant differences in profiles (*p* < 0.05, ANOVA); n.s. indicates 'not significant' (p > 0.05). **D**. Growth and fermentation capabilities of strains shown in B-C.

to the *bcy1Δ*, although its xylose fermentation capabilities were highly variable across replicates, for reasons we do not understand but could pertain to extremely high PKA activity. As such, the double mutant was not statistically different from either the Y184 parent or the *bcy1Δ* strain. Nonetheless, we used it to investigate the genetics of PKA signaling through these different branches. The three strains grow indistinguishably on glucose (p> 0.05, S1A Fig) with similar glucose consumption (p > 0.05, S1B Fig) and ethanol production (p > 0.05, S1C Fig), indicating that these phenotypes are specific to anaerobic xylose conditions.

We started by comparing transcriptomic responses to identify transcripts whose abundance across the strain panel correlates with growth or anaerobic xylose metabolism. Cells were grown in an anaerobic chamber to mid-log phase on rich medium with glucose as a carbon source (YPD) then switched to rich medium containing only xylose (YPX) for three hours, long enough for the *ira2Δ* strain to resume growing (Fig 2A). We performed short-read sequencing to measure changes in transcript abundance after the glucose-to-xylose shift. To understand strain responses, we compared transcript abundances across strains grown under each condition; we also compared the fold change in transcript abundance within each strain responding to carbon shift. There were major differences in expression comparing the strains

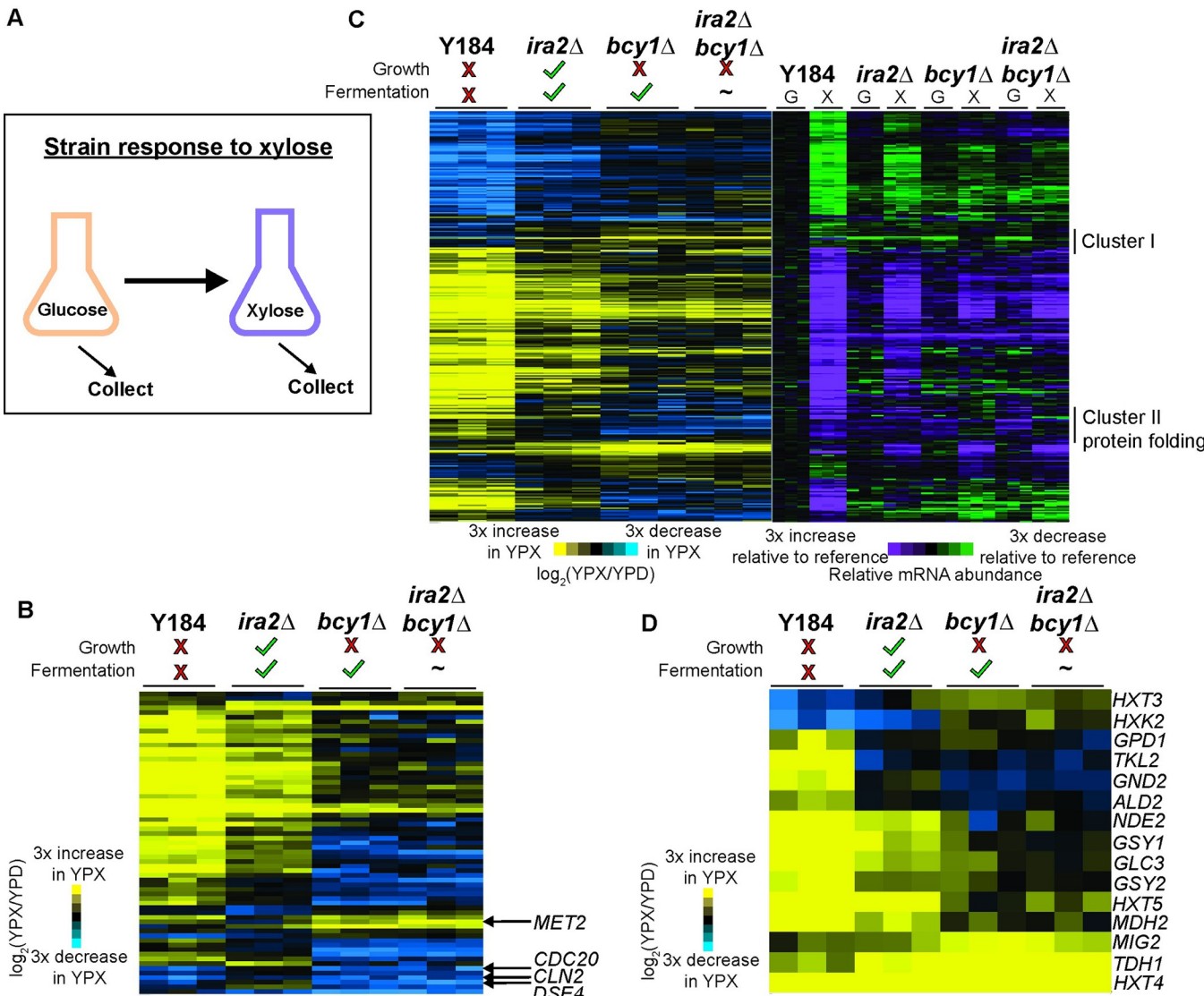

**Fig 2. Few transcriptomic patterns correlate with anaerobic xylose growth. A**. Experimental overview. Strains were grown anaerobically in rich glucose medium to early/mid-log phase, then switched to anaerobic rich xylose medium for three hours. **B**. Expression of 65 genes whose $\log_2$(fold change) upon glucose to xylose shift is different (FDR < 0.05) in at least one of the three non-growing strains (Y184, *bcy1Δ*, *ira2Δbcy1Δ*) compared to *ira2Δ*. Genes (rows) were organized by hierarchical clustering across biological triplicates measured for each strain (columns). Genes discussed in the text are annotated on the figure. **C**. Hierarchical clustering of 292 genes whose $\log_2$(fold change) upon glucose to xylose shift is different (FDR < 0.05) between the non-fermenting Y184 strain and the two robustly xylose fermenting strains (*ira2Δ*, *bcy1Δ*). The blue-yellow heatmap on the left represents the $\log_2$(fold change) in expression upon glucose to xylose shift across biological triplicates (columns). The purple-green heatmap on the right represents the abundance of each transcript (rows) in each strain grown on glucose (G) or xylose (X), relative to the average (n = 3) abundance of that transcript measured in the Y184 YPD sample. Clusters I and II are described in the text. **D**. Expression of 15 genes from C) that have annotations linked to glycolysis, gluconeogenesis, TCA cycle, and carbohydrate storage.

growing on xylose, whereas only mild expression differences were observed comparing strains grown on glucose (see Fig 2C, right panel). Correspondingly, strains do not differ substantially in their ability to grow anaerobically on glucose (S1 Fig) [35]. Thus, the differences in the fold-change expression response to the carbon shift are driven by differences in the xylose condition.

Our expectation at the outset was two-fold. On the one hand, we expected to find expression changes common to the xylose fermenting *ira2Δ* and *bcy1Δ* mutants, but discordant in

Y184 cells–these expression patterns may relate to xylose metabolism, since the Y184 strain is incapable of xylose metabolism [31,32]. On the other hand, expression patterns unique to the *ira2Δ* strain–the only strain capable of growing anaerobically on xylose–may reflect expression patterns related to growth.

### Few gene expression patterns correlate strictly with growth phenotypes

Somewhat surprisingly, there were few genes whose expression correlated strictly with growth phenotypes. Only two genes showed xylose-responsive expression changes that were specific to *ira2Δ* cells compared to the other three strains analyzed as a group in the statistical model (FDR < 0.05; see Methods): daughter-cell-specific glucanase *DSE4* and L-homoserine-O-acetyltransferase *MET2*. In fact, hierarchical clustering of all genes with a transcriptomic change in response to the carbon shift showed that the *ira2Δ* strain's response to xylose shift was most similar to that of Y184 cells, even though one strain can grow on and anaerobically ferment xylose and the other cannot (S2A Fig and S1 Table). We next performed pairwise comparisons of the glucose-to-xylose fold-change responses between each strain and the *ira2Δ* strain, then combined the lists of genes identified in all three comparisons. This method identified 65 genes; however, investigating the expression patterns once again indicated that the *ira2Δ* response was most similar to Y184 cells but with weaker magnitudes of change (Fig 2B and S2 Table). This set of 65 genes was enriched for genes induced in the environmental stress response (iESR genes [36], $p = 2 \times 10^{-7}$, hypergeometric test). Many genes induced in the Y184 and *ira2Δ* strains, but largely not in *bcy1Δ* strains, included genes related to metabolism, including several in the mitochondrial TCA cycle and peroxisomal fatty-acid oxidation pathway, which may reflect that *bcy1Δ* strains are more likely to recognize xylose as a fermentable carbon source. We specifically investigated the set of 65 genes for those that encode cell-cycle regulators and kinases, since these may be involved in growth kinetics; however only three, six, or eight genes within these categories were differentially expressed in Y184, *bcy1Δ*, or *ira2Δbcy1Δ* cells, respectively, compared to *ira2Δ* cells in response to xylose shift (FDR < 0.05). The *ira2Δ* strain showed weakly lower expression of cyclin *CLN2* and anaphase-promoting complex *CDC20*, whereas other strains showed strong reduction in expression (S3 Table). Transcript abundance for these genes is known to fluctuate during the cell cycle, thus while it is possible their expression influences growth arrest, it is likely that the expression of these genes reflects the expected difference between cycling (*ira2Δ*) and non-cycling (Y184, *bcy1Δ*, *ira2Δbcy2Δ*) cells. Expression of cell-cycle genes did not implicate arrest in a particular cell-cycle stage, consistent with early transcriptomic studies that showed that gene expression during cell-cycle arrest does not parallel expression of cells cycling through those phases [37].

Previous chemostat studies reported that repression of ribosomal protein (RP) and ribosome biogenesis (RiBi) genes is correlated with decreased growth, and these studies proposed that expression of these genes can predict cellular growth rate [38–41]. However, here we saw no correlation of RP and RiBi transcript abundance or response with growth phenotypes. The Y184 strain strongly repressed RP and RiBi genes upon xylose shift, which might be expected for a strain that arrests its growth, but so too did the *ira2Δ* strain, albeit with weaker magnitude of repression. Surprisingly, *bcy1Δ* and *ira2Δbcy1Δ* cells, whose growth is largely arrested after the xylose shift, showed little change in RP and RiBi transcripts compared to glucose-dependent growth (FDR <0.05, S2B Fig and S4 Table). These results reinforce past work from our lab that the expression of ribosome-associated genes does not necessarily parallel growth rate [42]. They further suggest that *bcy1Δ* strains cultured in xylose are unlikely limited by the abundance of RP and RiBi transcripts. Overall, while the non-growing strains have stronger

repression of a few cell-cycle regulators when compared to the *ira2Δ* strain, there was not a clear gene expression pattern to describe why *ira2Δ* cells grow and *bcy1Δ* strains do not.

## Few gene expression patterns correlate strictly with metabolism phenotypes

We next investigated shared gene expression changes related to robust anaerobic xylose fermentation. We compared expression in the Y184 strain responding to the xylose shift to the *ira2Δ* and *bcy1Δ* strains analyzed as a single group in the statistical model (we excluded the *ira2Δbcy1Δ* strain due to the variability of its fermentation phenotype, although it is capable of anaerobic xylose fermentation). This identified 292 differentially expressed genes (FDR < 0.05; S5 Table). Hierarchical clustering revealed that these genes typically had larger expression changes in Y184 and that those expression changes were progressively weaker across the strain series; once again, Y184 and the *ira2Δ* strain were more similar to one another than they were to the *bcy1Δ* strain (Fig 2C and S5 Table). Collectively, these genes were heavily enriched for genes in the ESR ($p = 3.624 \times 10^{-10}$, hypergeometric test).

Deeper interrogation revealed several small gene clusters of interest. Cluster I contained 19 genes induced in the *ira2Δ* and *bcy1Δ* strains but repressed in the Y184 strain. This group did not contain any functional enrichments; however, proteins encoded by several of these genes localize to the endoplasmic reticulum. Cluster II contained 31 genes induced in Y184 and either unchanged or repressed in both *ira2Δ* and *bcy1Δ* strains. This group was enriched for genes involved in protein folding ($p = 9.49 \times 10^{-6}$, hypergeometric test) and included the Hsp90 chaperone and cochaperone genes *HSP82*, *STI1*, and *AHA1*, as well as the mitochondrial matrix protein chaperone *HSP10*. Hsp90 can act as a signal transducer for alternative carbon source metabolism [43], again suggesting that Y184 does not recognize xylose as a fermentable carbon source.

We specifically interrogated the 292 genes for those involved in glycolysis, gluconeogenesis, TCA cycle, and carbohydrate storage, predicting that differences in expression would relate to altered xylose metabolism capabilities. This identified 15 genes with functional annotations linked to at least one of these processes (Fig 2D and S6 Table). The xylose fermenting strains shared expression at several hallmark genes. For example, Y184 cells strongly induced hexose transporter *HXT5*, normally induced by non-fermentable carbon sources, whereas the three mutant strains showed a weaker induction (FDR = $4.25 \times 10^{-5}$). The glucose-repressed aldehyde dehydrogenease *ALD2* was induced in Y184 and either did not change (*ira2Δ* cells) or was repressed (*bcy1Δ* cells) upon the switch to xylose (FDR = $4.32 \times 10^{7}$). Furthermore, glucose-induced transcriptional repressor *MIG2* showed stronger induction in the xylose fermenting strains, and especially *bcy1Δ* strains, compared to Y184 (FDR = 0.047). These data are all consistent with the hypothesis that the xylose fermenters recognize xylose as a fermentable carbon, whereas Y184 activates a carbon-starvation response.

## Regulatory analysis reveals strain-specific differences in carbon, iron, and lipid gene control

We next focused on understanding how growth and metabolism are decoupled in the *bcy1Δ* strain, and we thus directly compared its expression to that in *ira2Δ* cells. We focused on genes whose expression changes in response to the xylose shift were in opposing directions to implicate processes involved in decoupling growth and metabolism (Fig 3A and S7 Table, see Methods). Among the identified genes, we scored enrichment of functional terms as well as known targets of transcriptional regulators (S8 Table). We also used motif analysis to discover shared sequence motifs upstream of genes uniquely induced or repressed in the *bcy1Δ* strain, and

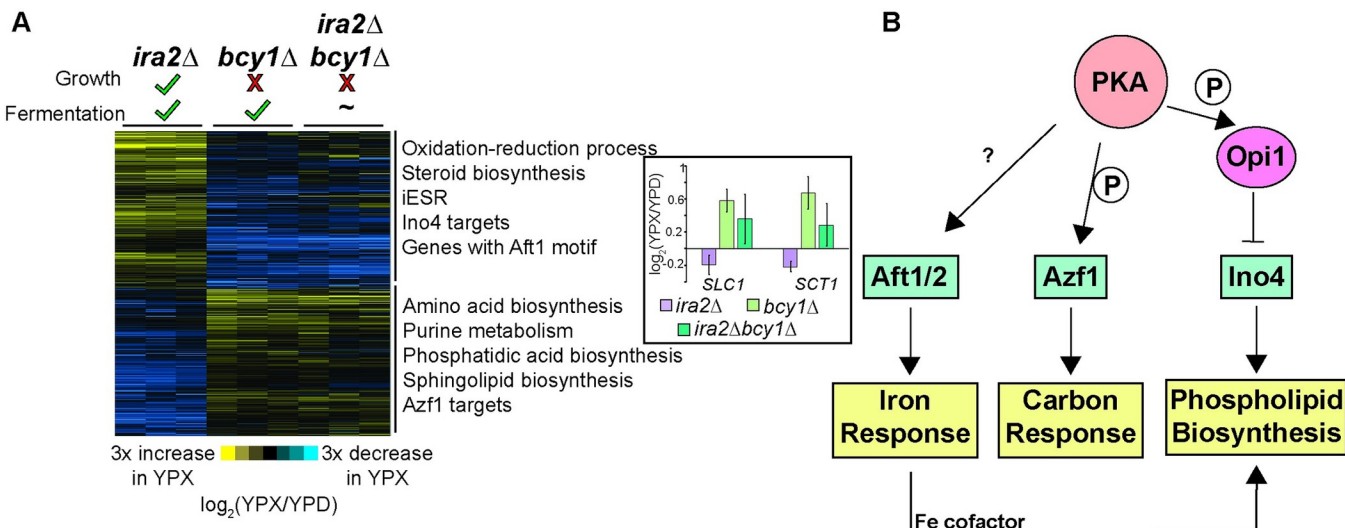

**Fig 3. Genes uniquely expressed in the *bcy1Δ* strain implicate an integrated response to xylose metabolism and growth coupling. A**. Expression of 654 genes whose log$_2$(fold change) upon glucose to xylose shift is different (FDR < 0.05) between the *ira2Δ* and *bcy1Δ* strains and whose expression change is in the opposite direction across the two strains (see Methods for details). Significant functional enrichments are annotated next to the two main clusters ($p < 10^{-4}$, hypergeometric test). Bar graph inset represents the log$_2$(fold change) of the two phosphatidic acid biosynthesis enzymes in this group, see text for details. **B**. Regulatory relationships between transcription factors whose targets or known binding sites were enriched in (**A**). Documented PKA-dependent phosphorylation is indicated by a P. See text for details.

then matched those to known transcription factor binding sites (see Methods). We identified 654 genes differentially expressed in *bcy1Δ* cells and with a fold-change in the opposite direction as *ira2Δ* cells upon the glucose-to-xylose shift (Fig 3A and S7 Table). Importantly, only 82 genes (12.5%) showed significant differences in basal gene expression when cells were grown on glucose (S3A Fig), indicating that the majority of genes are identified due to differences in response to xylose shift.

The results implicated several regulators, some with prior connections to anaerobic xylose fermentation. 318 genes induced in the *bcy1Δ* strain shifted to xylose, but repressed in the *ira2Δ* cells, were enriched for amino acid and sphingolipid biosynthesis genes, as well as targets of the carbon-responsive Azf1 transcription factor ($p < 10^{-4}$, hypergeometric test). Previous work from our lab implicated Azf1 in anaerobic xylose fermentation, and indeed, we showed that the over-expression of *AZF1* in an *ira2Δ* strain enhances the rate of anaerobic xylose utilization [32]. Additionally, PKA has been implicated in Azf1 phosphorylation [44]; together with the fact that the *AZF1* gene is uniquely induced in the *bcy1Δ* strain suggest its functional importance in xylose metabolism (see Discussion).

In contrast, several regulators were implicated by the 336 genes uniquely repressed in the *bcy1Δ* strain. These included genes harboring upstream binding sites of the iron-responsive Aft1/2 transcription factors (S3B Fig) and known targets of transcriptional activator Ino4 that responds to inositol for phospholipid biosynthesis (Fig 3A; see more below). Iron is an important cofactor of many enzymes, including those involved in mitochondrial respiration, lipid biogenesis, and amino acid biosynthesis, all of whose genes were among the differentially regulated genes studied here. Additionally, Aft1/2 regulation and the iron regulon have been linked with PKA activity; however, direct interactions remain to be identified [45]. Interestingly, Aft1/2 and Azf1 both are both connected to the regulator Mga2, which controls lipid and hypoxia genes and that we previously showed enhances anaerobic xylose fermentation when over-expressed in *ira2Δ* cells [32,46] (see Discussion). Targets of the Ino2/4 regulators that respond

to inositol for phospholipid biosynthesis were also present in this gene set; while a majority of the targets identified here were repressed in the *bcy1Δ* strain, some of the known targets were repressed in the *ira2Δ* strain but induced in the *bcy1Δ* mutant (S3C Fig and S9 Table). This may reflect the complexities of the genes' regulation by other factors. Nonetheless, Ino2/4 targets were enriched among the genes oppositely regulated in the *bcy1Δ* versus *ira2Δ* strain. Overall, these results provide an interesting link between PKA signaling, carbon and iron responses, and lipid metabolism (Fig 3B).

The presence of many lipid biosynthesis genes in this gene set and the highly regulated role of lipids in cell growth and proliferation prompted a deeper investigation of lipid metabolism genes. The *bcy1Δ* strain repressed genes involved in ergosterol biosynthesis and some targets of Ino2/4 that are involved in phospholipid metabolism (Figs 3A and S3C). This response is consistent with the model that Ino2/4 activity is reduced. However, the *bcy1Δ* strain also induced some genes involved in the synthesis of phosphatidic acid (PA) (Fig 3A inset), which normally promotes Ino4 activity by sequestering Ino4's inhibitor Opi1 to the ER membrane [47]. This response suggests that some connection between PA, Opi1, and Ino4 is disrupted in the absence of *BCY1*. PKA is known to regulate the Ino2/4 pathway through direct phosphorylation of Opi1 to increase its inhibitory activity [48]. Together, these results raised the possibility that the *bcy1Δ* strain has important differences in lipid metabolism and perhaps composition, which could be modulated by differences in PKA activity in this strain.

## Lipidomic and phosphoproteomic analyses show disrupted phospholipid metabolism in bcy1Δ cells

Since the transcriptomic responses implicated differences in lipid metabolism, we investigated the lipidomic composition of our strains. Strains were grown in a similar design as the transcriptomic analysis, where anaerobically glucose-grown Y184, *ira2Δ*, *bcy1Δ*, and *ira2Δbcy1Δ* cells were shifted to anaerobic xylose media for three hours before lipids were analyzed by mass spectrometry (see Methods). We detected over 4000 lipid species including 239 that were confidently assigned to a particular lipid class (S10 Table). All detected lipid species were included in the statistical analysis to obtain a wholistic understanding of lipidome differences between the strains. We again sought to find lipidomic profiles correlated with xylose metabolism and growth, xylose metabolism but no growth, and no xylose metabolism or growth.

We compared the Y184 strain to the three strains with upregulated PKA activity and identified 18 lipids whose change in abundance upon a shift to xylose significantly differed in Y184 cells. This group included phosphatidylserine (PS) species (Fig 4A and S11 Table). Interestingly, all three mutants increased the abundance of these PS species when shifted to xylose, whereas Y184 cells decreased the abundance of one and failed to induce the other to the same degree as the mutants. The gene encoding the PS synthase *CHO1* was strongly induced in Y184 cells, indicating that the decrease in PS in Y184 cells is unlikely due to decreased *CHO1* expression. Instead, we analyzed previous phosphoproteomic data from our lab and discovered that Cho1 was phosphorylated to a much higher degree in the Y184 strain on serine 46 ($|\log_2\text{FC}| > 1$, Table 1), a known PKA site that inhibits Cho1 activity [49]. Together, these results indicate PKA-dependent inhibition of PS synthesis in Y184 cells.

We next compared lipidomic profiles in the growing *ira2Δ* strain shifted to xylose to the *bcy1Δ* and *ira2Δbcy1Δ* strains that do not grow. Due to limited statistical power (caused by replicate variation in one of the three *bcy1Δ* strain replicates), we compared the *ira2Δ* response to *ira2Δbcy1Δ* cells, whose response was highly similar to two out of the three *bcy1Δ* strain replicates. One caveat of this analysis is that the *ira2Δbcy1Δ* strain displays a variable anaerobic-xylose fermentation profile; nonetheless, given the similarity to *bcy1Δ* phenotypes, the high

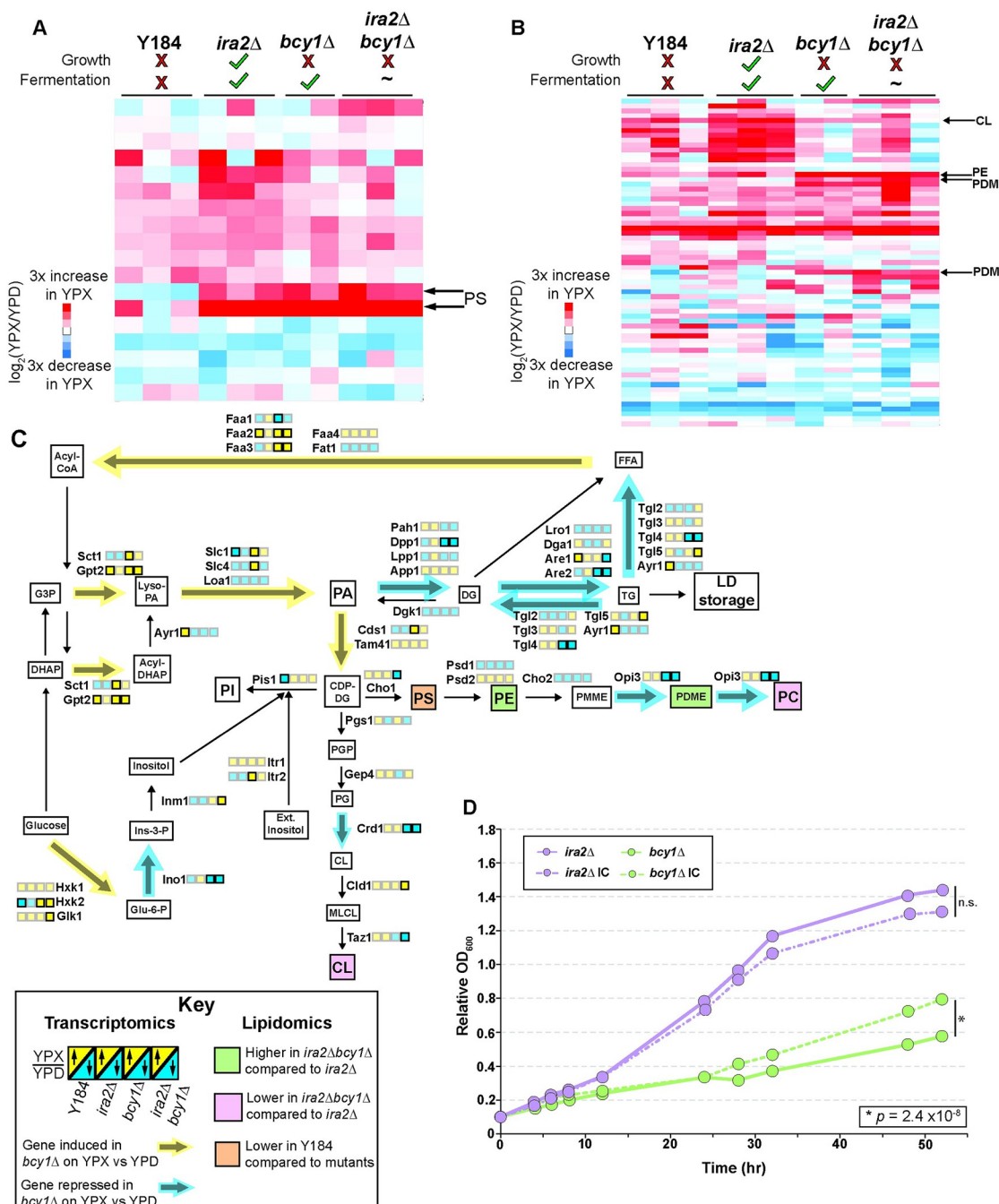

**Fig 4. *bcy1Δ* strains show altered phospholipids after anaerobic xylose shift. A-B.** Abundance of lipids (rows) with a significant difference in $\log_2$(fold change) upon anaerobic glucose-to-xylose shift in **(A)** Y184 compared to PKA pathway mutants (*ira2Δ*, *bcy1Δ*, *ira2Δbcy1Δ*) analyzed as a group in the statistical model or **(B)** *ira2Δ* cells compared to *ira2Δbcy1Δ* cells. Lipids of interest are annotated. **C.** Partial phospholipid biosynthesis pathway with transcriptomic and lipidomic data represented. Yellow-blue boxes next to each enzyme name represent the average $\log_2$(fold change) in transcript abundance upon glucose-to-xylose shift for each strain, as outlined in the key. Significant differences compared to the *ira2Δ* strain (FDR < 0.05) are represented in sharp, bolded boxes, whereas insignificant differences are translucent. Colorized pathway arrows (yellow: induced, blue: repressed) represent the predominant transcript patterns for that enzymatic step when comparing the *bcy1Δ* and *ira2Δ* strains. Lipids whose fold-change in abundance is different in specific strains are according to the key. Lipid abbreviations: FFA–free fatty acids; PA–phosphatidic acid; DG–diacylglycerol; TG–triacylglycerol; PI–phosphatidylinositol; PS–phosphatidylserine; PE–phosphatidylethanolamine; PMME–monomethyl-phosphatidylethanolamine; PDME–dimethyl-phosphatidylethanolamine; PC–phosphatidylcholine; CL–cardiolipin. **D.** Average (n = 4) change in $OD_{600}$ of *ira2Δ* and *bcy1Δ* grown anaerobically in rich xylose medium either in the absence (solid lines) or presence (dashed lines, IC) of inositol (75 μM) and choline (10 mM) (* indicates $p = 2.4 \times 10^{-6}$, ANOVA).

**Table 1. Phosphorylation changes of phospholipid biosynthetic enzymes.**

| Protein | Residue | Av. logFC (*bcy1Δ-ira2Δ*) | Av. logFC (*ira2Δbcy1Δ-ira2Δ*) | Av. logFC (Y184-*ira2Δ*) | Av. logFC (Y184-*bcy1Δ*) |
|---------|---------|---------------------------|--------------------------------|--------------------------|--------------------------|
| Are1 | S45 | 0.445 | 0.695 | 0.635 | 0.19 |
| Cho1 | S46 | -0.35 | -0.37 | 1.08 | 1.43 |
| Hxk1 | Y270 | 1.45 | 1.41 | 0.74 | -0.71 |
| Hxk1 | S262 | 1.57 | 1.28 | 1.1 | -0.47 |
| Hxk1 | S293 | 0.54 | 0.1 | 1.44 | 0.9 |
| Hxk1 | S158 | -1.3 | -0.74 | 0 | 1.3 |
| Hxk2 | S385 | -0.65 | -0.68 | 1.17 | 1.82 |
| Hxk2 | S158 | -0.58 | -0.38 | 0.01 | 0.59 |
| Pct1 | T59 | 0.19 | 0.94 | -0.24 | -0.43 |
| Pah1 | S166 | -0.52 | -0.5 | 0.3 | 0.82 |
| Pah1 | S823 | -1.53 | -1.49 | 0.84 | 2.36 |
| Pah1 | S168 | -0.33 | -0.44 | 0.3 | 0.62 |
| Are2 | S176 | 0.78 | 0.44 | 0.63 | -0.15 |
| Slc4 | S512 | 0.58 | 1.01 | 1.01 | 0.42 |

reproducibility of the double mutant's transcriptomic and lipidomic profiles suggests a good representation of hyper-active PKA signaling. It is possible that this analysis may miss some lipidomic changes related to the variation in *ira2Δbcy1Δ* metabolism profiles. Even so, we identified 67 lipids whose fold-change was significantly different in *ira2Δbcy1Δ* cells upon xylose shift versus *ira2Δ* cells (FDR < 0.05, Fig 4B and S12 Table). The analysis confidently classified six of the lipids, including phosphatidylethanolamines (PE), phosphatidyl dimethylethanolamines (PDME), and cardiolipins (CL).

PE and multiple PDME species were more abundant in the *ira2Δbcy1Δ* strain exposed to the shift compared to *ira2Δ* cells (FDR < 0.05, Fig 4B). These differences were particularly interesting because PE is further metabolized to PDME and then to phosphatidylcholine (PC), the most abundant phospholipid in the cell, through three consecutive methylation reactions by Cho2 and Opi3, respectively (Fig 4C) [50]. While the *CHO2* transcript was not differentially expressed between *ira2Δ* and *bcy1Δ* strains, *OPI3* was: *ira2Δ* cells shifted to xylose induced *OPI3* expression, whereas *bcy1Δ* and *ira2Δbcy1Δ* cells repressed it (FDR = $2.45 \times 10^{-12}$ and FDR = $6.22 \times 10^{-13}$, respectively). Previous studies suggest that blocking PC synthesis through *OPI3* deletion, but not *CHO2* deletion, inhibits growth due to the accumulation of phosphatidyl monomethylethanolamine (PMME) and insufficient PC production [51]. To investigate effects on PC, we analyzed all PC lipid moieties in the dataset; PC lipids were reproducibly lower in abundance after the xylose shift in *bcy1Δ* cells when compared to *ira2Δ* cells (*p* = 0.000419, ANOVA; S4 Fig and S13 Table). We propose that the *bcy1Δ* strain experiences a bottleneck in that pathway leading to PC synthesis from PE, which may impact its ability to grow on xylose (see Discussion).

Among other lipids whose abundance was influenced by *BCY1* deletion and xylose shift was cardiolipin, a major component of mitochondrial membranes critical for a variety of functions including acetyl coA synthesis, TCA cycle, iron metabolism, arginine metabolism, and protein import [52]. Interestingly, cardiolipin abundance was reduced in the *ira2Δbcy1Δ* strain upon xylose shift compared to *ira2Δ* cells. The difference is underscored by transcriptomic differences, since several cardiolipin biosynthetic genes were induced in *ira2Δ* cells but repressed or induced to a weaker extent in *bcy1Δ* and *ira2Δbcy1Δ* strains (FDR < 0.05). Additionally, production of PS, PE, and PC is dependent on properly functioning mitochondrial membranes as PS is shuttled into the mitochondria and converted to PE by the phosphatidylserine

decarboxylase Psd1, before PE is shuttled back to the ER. Thus, the effects of cardiolipin reduction in *bcy1Δ* strains are further compounded by impacting other branches of phospholipid biosynthesis.

We expected to see differential abundance of PA in *ira2Δbcy1Δ* cells versus *ira2Δ* cells, since *bcy1Δ* and *ira2Δbcy1Δ* strains induced some PA biosynthesis genes whereas *ira2Δ* cells do not (Fig 3A). While there were no significant differences in PA moieties between the strains (FDR > 0.05), we did identify altered phosphorylation status of the PA phosphatase enzyme Pah1 (S823; Table 1). Pah1 converts PA to diacylglycerol, which is funneled into storage lipids [50]. Phosphorylation of serine 823 is significantly lower in the *bcy1Δ* and *ira2Δbcy1Δ* strains compared to the *ira2Δ* strain ($\log_2$(fold change) < -1). Interestingly, this serine has not been previously annotated as a phosphorylated residue (BioGRID version 4.4.213) [53], but it is within a potential PKA consensus site (RRxxS/T). PKA is known to phosphorylate Pah1 at another residue not captured in our dataset to inhibit its activity [54]. Our results raise the possibility that S823 regulates Pah1 activity in a manner that affects PA in these strains. Overall, the differences seen in PE, PDME, and PC abundances, as well as differences in transcript abundance and phosphorylation status of phospholipid biosynthesis enzymes, suggest a bottleneck in the pathway in the *bcy1Δ* strains that may inhibit their ability to proliferate on xylose (see Discussion).

## Supplementation with phospholipid precursors only modestly improves growth

We questioned if supplementing xylose medium with phospholipid precursors, particularly inositol and choline that can be funneled into phospholipid biosynthesis via the Kennedy Pathway, may bypass a possible bottleneck and thus rescue the *bcy1Δ* strain's growth. We therefore grew *bcy1Δ* and *ira2Δ* strains anaerobically in xylose medium with and without choline and inositol supplementation (we included inositol since the *INO1* gene is repressed in *bcy1Δ* cells) (Fig 4C and S5 Table). After 52 hours of growth in supplementation, *bcy1Δ* cells experienced a very modest but statistically significant growth improvement ($p = 2.4 \times 10^{-6}$, ANOVA; Fig 4D), whereas the *ira2Δ* strain did not. While the *bcy1Δ* strain's inability to grow anaerobically on xylose cannot be fully explained by a deficiency in phospholipid precursors, the modest improvement implicates it as a contributing factor to the phenotype.

## Growth and metabolism can be genetically recoupled through directed evolution

We took a second approach to identify pathways and processes responsible for growth coordination in *bcy1Δ* strains by conducting adaptive laboratory evolutions to recouple xylose-dependent growth and metabolism. The *bcy1Δ* strain was first grown anaerobically in rich medium supplemented with 2% glucose to accumulate mutations [55], then the culture was seeded into fresh anaerobic medium containing 2% xylose and 0.1% glucose and passaged periodically for ~35 generations until the culture showed robust changes in cellular density over time (see Methods). Single colonies were isolated and characterized for their growth and fermentation capabilities, and genetic changes were identified through whole genome sequencing (see Methods). Three independent evolutions were performed, and several colonies were selected at different stages of the evolutions.

In all three experiments, we identified mutants with recoupled growth and metabolism despite the absence of *BCY1*, evident by their robust anaerobic growth on xylose medium compared to the *ira2Δ* strain (Figs 5A, S5 and S6). Interrogating the genome sequences identified multiple mutations in each strain, along with copy-number variations and aneuploidy in

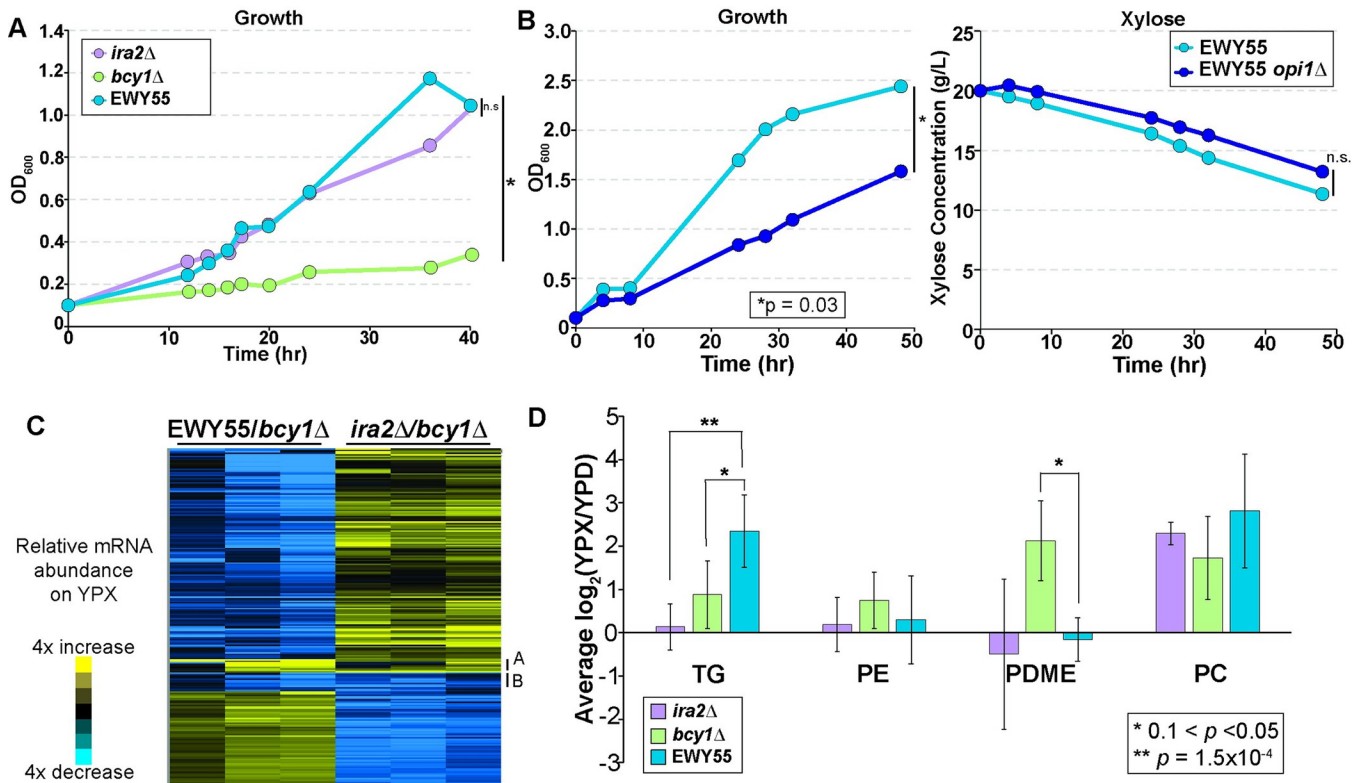

**Fig 5. Directed evolution recoupled growth and metabolism on xylose. A**. Average (n = 3) change in OD$_{600}$ of *ira2Δ*, *bcy1Δ*, and EWY55 strains grown anaerobically in rich xylose medium (*, $p < 10^{-4}$, ANOVA; n.s., not significant). **B**. Change in OD$_{600}$ (left panel) and xylose concentration (right panel) over 48 hours of EWY55 and EWY55 *opi1Δ* strains grown anaerobically on rich xylose medium. (*, $p < 0.05$, ANOVA). **C.** Expression of 233 genes whose transcript abundance during growth on xylose was significantly different in EWY55 and/or *ira2Δ* strains compared to the *bcy1Δ* strain (FDR < 0.05), visualized by hierarchical clustering. Data represent the log$_2$ transcript abundance in each strain grown anaerobically in xylose compared to *bcy1Δ* strain. Cluster A (9 genes) and B (13 genes) are annotated, see text for details. **D.** Bar plot of the average and standard deviation log$_2$(fold change) (n = 3) in lipid abundance of key lipids with reproducible differences 1.5-fold or greater in EWY55 compared to *ira2Δ* or *bcy1Δ* strains. Asterisks denote significant differences by ANOVA.

several of the evolved lines (Table 2). Only evolved mutations impacting the coding sequence of a gene were analyzed further. Interestingly, there was no genetic change common to all evolved strains, strongly suggesting multiple routes to recoupling growth and metabolism in the absence of *BCY1*. Four of the characterized strains from the three experiments regained growth rates comparable to and statistically indistinguishable from *ira2Δ* cells ($p > 0.05$,

**Table 2. Genetic changes in evolved *bcy1Δ* strains.**

|  | Gene | Nucleotide change | Amino acid change | Chromosome Duplications |
|---|---|---|---|---|
| EWY55 | *OPI1* | T715G | S239A | Chr. I |
|  | *TPK1* | G829C | A277P |  |
|  | *TOA1* | T354A | N118K |  |
|  | *RIM8* | C1591T | Q531* |  |
| EWY87-1 | - | - | - | Chr. VIII |
| EWY87-3 | *RPA43* | A751G | S251N | - |
| EWY89-1 | *HSC82* | G235C | D79H | Chr. X, XVI |
| EWY89-2 | - |  |  | Chr. I, X, XVI, XIV$^{560217-625584}$ |
| EWY89-3 | *HSC82* | G235C | D79H | Chr. I, IX, X, XVI, XIV$^{560217-625585}$ |

ANOVA), including EWY55 from the first culture, EWY87-1 and EWY87-3 from the second culture, and EWY89-3 from the third evolution culture (S6A and S6B, S6E Fig). Strains EWY89-1 and EWY89-2 showed modest growth on xylose but did not differ significantly from the *bcy1Δ* strain ($p > 0.05$, ANOVA; S6C and S6D Fig). Genetic changes for all evolved strains are listed in Table 2.

Strain EWY55 was particularly interesting. This strain harbored nonsynonymous mutations in several genes, including PKA catalytic subunit *TPK1*, the negative regulator of phospholipid genes, *OPI1*, described above, *RIM8* that is required for anaerobic growth [56], and TFIIA large subunit *TOA1* (Table 2). The *OPI1* mutation was especially interesting because Opi1 was implicated in the phospholipid transcriptomic analysis above (Fig 3) and because the mutation changes a known phosphorylation site, serine 239, to alanine (Table 2). CKII has been reported to phosphorylate this site and was previously shown to activate Opi1 [57]. This poses the question of whether Opi1 is aberrantly regulated in the *bcy1Δ* strain, and if this is responsible for its lack of growth on xylose.

To identify causal alleles responsible for recoupling growth and metabolism in the EWY55 strain, we performed single gene deletions and allele swaps in the *bcy1Δ* and EWY55 strains (see Methods). This strain background is derived from a wild isolate that is less genetically amenable than laboratory strains [58], and we were unable to recover *TPK1* deletion in either strain despite many efforts. Deletion of *RIM8* or *TOA1* did not impact the growth of EWY55 cells, nor did substituting the parental alleles into the evolved strain (S5C Fig). However, deletion of the evolved *opi1* gene partially but significantly reduced anaerobic xylose growth of the EWY55 strain in liquid medium (Fig 5B). Importantly, the strain retained robust xylose fermentation, indicating that Opi1 plays a role in the coupling of growth and metabolism (see Discussion). Complementation experiments to swap strain alleles were not successful, since introducing even the empty vector into this strain complemented anaerobic xylose growth on a plate for reasons that are not clear but may suggest that the cells grow differently during drug selection (S5C Fig). While we cannot be sure *OPI1* is the causal gene, our results indicate that the genetics modulating this trait is complex and may result from different evolutionary paths, but at least in EWY55 is likely to include a role for evolved Opi1 function.

## Transcriptomic and lipidomic analysis in the evolved strain reveals altered phospholipids

To further characterize the evolved EWY55 strain, we performed another transcriptomic and lipidomic experiment as described above (see Methods) with the main goal of identifying if the evolved EWY55 strain reverted its gene expression and lipid composition to that of the *ira2Δ* strain. Surprisingly, the EWY55 strain did not recapitulate the *ira2Δ* gene expression or lipid abundance profiles at most entities. We identified 297 transcripts less abundant in EWY55 growing anaerobically on xylose compared to the *bcy1Δ* strain (FDR < 0.05; S14 Table), and these were enriched for genes involved in mitochondrial functions, such as electron transport chain, oxidation-reduction, and targets of the HAP2/3/4/5 complex; genes involved in phospholipid metabolism; and genes involved in ergosterol synthesis ($p < 0.05$, hypergeometric test, see Methods). Many of these processes were significantly affected in our original comparison of the *bcy1Δ* and *ira2Δ* strains. Additionally, 93 genes with higher abundance in the EWY55 compared to the *bcy1Δ* cells (FDR < 0.05; S14 Table) were enriched for ribosomal protein genes and genes involved in translation and sulfate assimilation ($p < 10^{-7}$, hypergeometric test), processes important for rapid growth. Since EWY55 cells recapitulated the xylose-dependent growth seen in *ira2Δ* cells, we next asked if its expression changes recapitulated *ira2Δ* patterns relative to *bcy1Δ* cells–surprisingly, most did not (Fig 5C). This indicates that

the evolved EWY55 did not recouple growth and metabolism under anaerobic xylose conditions via reverting to the *ira2Δ* strain's expression patterns. There were a few exceptions, including 22 transcripts of diverse functions (Fig 5C, Clusters A and B) in which expression differences in EWY55 recapitulated those seen in *ira2Δ* cells compared to the *bcy1Δ* strain (S15 Table; FDR < 0.05). While the role of these expression changes will require future study, it is intriguing that these clusters included several targets of the glucose-responsive transcription factor Rgt1 and the Sok2 regulator that responds to starvation and hypoxia; both genes have connections to PKA signaling [59–62]. We were particularly interested in phospholipid biosynthesis genes, given all the connections to this pathway throughout our studies. In general, EWY55 cells showed lower transcript abundances of phospholipid biosynthesis genes compared to the *bcy1Δ* strain grown anaerobically on xylose (S16 Table), making its expression even more divergent from the *ira2Δ* strain.

The phospholipid composition further supports the unique changes of the EWY55 strain that permit recoupled growth and metabolism on xylose. The EWY55 strain showed significantly greater abundance of the storage lipid triacylglycerol (TG; Fig 5D and S17 Table; $p < 10^{-4}$, ANOVA). Importantly, EWY55 had significantly lower levels of PDME and trended towards higher levels of PC compared to *bcy1Δ* cells, recapitulating the pattern in *ira2Δ* cells (Fig 5D and S17 Table). These results are consistent with the hypothesis that the evolved EWY55 strain altered the pathway compared to *bcy1Δ* cells. Together, our results underscore the complexity of responses to xylose growth and metabolism across *ira2Δ* cells, the parental *bcy1Δ* strain, and EWY55 (see Discussion).

## Discussion

We began this work with two primary goals: to identify signatures of xylose-dependent growth and metabolism across a suite of strains with varying capabilities and to elucidate the mechanism through which growth and metabolism are decoupled in cells lacking *BCY1*. One key result from our work is that there is no obvious gene expression signature associated with the ability to grow anaerobically on xylose (Fig 2). While we did identify a handful of cell-cycle genes whose expression was consistent with cycling in the *ira2Δ* strain, there were no clear signatures correlated with growth. This was especially interesting in the case of ribosome-related genes, since there has been much debate about whether the level of RP transcripts underlies growth rate [38–42]. In chemostat experiments where growth is limited by nutrient restriction, the abundance of RP and RiBi genes correlates with growth rate, consistent with one set of long-standing models of growth limitations in bacteria [63–68]. However, other seminal studies focusing on stress conditions suggest that growth during stress is not limited by ribosome production [42,69–73]. Our results show clearly that expression of RP and RiBi genes is higher in the non-growing *bcy1Δ* strains than dividing *ira2Δ* cells (S2B Fig). In contrast, the EWY55 strain that recovers anaerobic growth on xylose shows higher expression of ribosome-related genes, perhaps supporting rapid division. Together, our results add to a growing body of work that shows that, although production of ribosome components is often correlated with growth rate, division dynamics cannot be universally predicted by RP and RiBi transcript abundances.

However, transcriptomic patterns did implicate an interconnected network of expression differences specific to the *bcy1Δ* strain, and in turn the evolved EWY55 strain, connected to PKA signaling (Fig 3). The affected network implicates mitochondrial function, iron response, carbon metabolism, and phospholipids. We propose that these processes are normally coordinated by PKA signaling in a manner that requires the regulatory subunit Bcy1. Our results are consistent with prior implications that these processes are involved in anaerobic xylose fermentation. Targets of the carbon-responsive transcription factor Azf1 were altered in the

*bcy1Δ* strain upon xylose shift compared to the *ira2Δ* strain (Fig 3A). We previously showed that altering expression of this transcription factor affects xylose fermentation rates and growth in an *ira2Δ* strain, and this was surprisingly connected to the ER-localized transcription factor Mga2 [32]. While Mga2 targets were not statistically enriched in comparisons here, 42% of the genes whose promoter is bound by Mga2 (11/26 genes) differed in expression between *bcy1Δ* and *ira2Δ* cells. Additionally, Aft1/2 activity and localization is dependent on Mga2 presence, thus adding another possible connection for Mga2 in our proposed regulatory network [74] (Fig 3B). Our results here strongly suggest that deletion of *BCY1* naturally augments the transcription factors' abundance and/or activity. These factors may indirectly alter mitochondrial and/or iron homeostasis. In fact, deletion of the iron-sulfur scaffold protein *ISU1*, an important sensor of iron availability, is required for anaerobic xylose metabolism [26,31]. Why *ISU1* deletion is required for xylose fermentation remains unclear, but one possibility is that it aids in metabolic rewiring influenced by the iron regulon, the Aft1/2 transcription factors, and altered levels of PKA activity [45,75–77].

Remarkably, PKA is directly connected to all these processes. Past work implicated PKA in directly phosphorylating Azf1 [32,44]. While a direct link between PKA and Aft1/2 activity has yet to be identified, PKA catalytic subunit Tpk2 is required to repress the high-affinity iron uptake pathway under standard conditions [76]. Additionally, Ira2 can localize to mitochondria, suggesting that PKA can also localize to this organelle [78]. In fact, PKA is found at the mitochondria of higher eukaryotes [79], suggesting that yeast PKA may also localize to the mitochondria. Together, our results suggest that upregulated PKA activity is required for xylose fermentation and can occur via either *IRA2* or *BCY1* deletion, but deletion of *BCY1* produces stronger effects that underscore its higher per-cell rate of xylose fermentation [32].

A fundamental aspect of *BCY1* deletion is that cells can no longer grow robustly despite enhanced xylose metabolism. Our integrated analysis points to a defect in phospholipid flux or metabolism as a major contributor to this decoupling. First, the *bcy1Δ* strains showed altered gene expression, including Ino2/4 targets such as *INO1* (Figs 3A, 4C and S3C), that pointed to differences in phospholipid metabolism. Second, we found that *bcy1Δ* strains grown anaerobically on xylose display an altered lipid profile that implicates altered metabolism in the PE-PD-ME-PC pathway, along with phosphorylation differences on key phospholipid enzymes (Table 1). Finally, re-evolving a coupling between anaerobic-xylose growth and metabolism in the *bcy1Δ* parent implicated mutations in PKA subunit *TPK1* and the Ino2/4 repressor *OPI1* (Table 2), which is known to be directly regulated by PKA phosphorylation [48]. Opi1 has complex roles in regulating phospholipids, including during the switch to invasive growth depending on nutrients [80], a process also regulated by the RAS/PKA pathway [81–85]. While we were unable to elucidate the exact role of these alleles, our results suggest that the *OPI1* mutation may alter Opi1 regulation, especially given that the identified mutation in Opi1 occurs at a known CKII kinase site that regulates Opi1 activity [57]. Our past network inference across this panel of engineered strains revealed altered phosphorylation of CKII targets [32]. Finding that complete deletion of the mutated *OPI1* allele reduced growth of the evolved EWY55 strain on xylose (Fig 5B) suggests altered Opi1 activity in the *bcy1Δ* strain is somehow resolved by mutation of this CKII site. We propose that an interplay between PKA and possibly CKII affect Opi1 regulation in the *bcy1Δ* strain, and that this interplay is important for growth coupling.

Importantly, phospholipid metabolism is required for growth and division. Cells must generate enough phospholipids to support membrane biogenesis [4,86–91]. Furthermore, phospholipids function in inter-organelle communication, connecting the ER and mitochondria via the ER-mitochondria encounter structure (ERMES). Impairment of this structure and inter-organelle communication is known to cause diverse mitochondrial phenotypes and disrupt phospholipid biosynthesis [92,93], connecting phospholipid metabolism to mitochondrial

functions, including xylose flux [31]. One possibility is that impaired regulation of Opi1 and Ino2/4 in *bcy1Δ* cells disrupt growth in the *bcy1Δ* strain due to insufficient levels of growth-supporting lipids (PC) via decreased production or increased recycling. But another possibility is that accumulation of methylated PE intermediates during the conversion to PC create a toxic buildup coupled with insufficient PC (Fig 4C). Ishiwata-Kimata et al. (2022) [51] found that accumulation of PMME leads to a growth defect by triggering the unfolded protein response and growth arrest. Accumulation of PDME in the *bcy1Δ* strain (Fig 4B and 4C) may also lead to ER stress, preventing growth paired with interfered ER-mitochondrial communication. Importantly, the evolved EWY55 strain does not share the *bcy1Δ* strain's accumulation of PMDE, leading us to propose that EWY55 cells have overcome the possible bottleneck in PC synthesis (Fig 5D). Future studies analyzing pathway flux are needed to fully confirm the presence of and recovery from a bottleneck in phospholipid biosynthesis.

A major remaining question is how deletion of *BCY1*, but not *IRA2*, decouples growth from metabolism specifically under the conditions studied here. One possibility is that *BCY1* deletion upregulates PKA activity to a higher level than deletion of *IRA2*, whose activation of PKA is indirect via cAMP regulation [33]. PKA activity over some threshold could cause decoupling, as deletion of *BCY1* is well characterized to sensitize cells to environmental stressors [94]. An alternate model is that localized cAMP production could influence when and where PKA is active in *ira2Δ* cells. cAMP exists in concentration gradients in cells to control the subcellular location of active PKA [95–97]. It is possible areas with low cAMP concentration locally inactivate PKA in the *ira2Δ* strain, whereas *BCY1* deletion leads to wholesale activation of PKA throughout the cell. Fitting with this model, *BCY1* deletion inhibits growth and metabolism on non-fermentable carbon sources, causing cell death during the diauxic shift and stationary phase, likely from uninhibited PKA [98,99]. However, a third possibility is that loss of *BCY1* leads to misdirection of PKA activity. PKA can be directed to subcellular targets in higher eukaryotes via A-kinase anchoring proteins (AKAP) that bind to and direct localization of PKA [100]. While yeast do not possess orthologs of AKAPs, functional analogs have been proposed including Bcy1 itself [79,101,102]. Anaerobic xylose growth and metabolism may be decoupled in *bcy1Δ* strains via disrupted subcellular localization and substrate interactions of PKA that are coordinated by Bcy1. Additionally, Bcy1 is reported to interact with fatty acid synthases subunits (Fas1/2) [102], implying a direct, physical connection between PKA and lipid biosynthesis. While future studies of PKA localization and substrate interactions are needed to confirm this model, our results show that Bcy1 plays a special role in coordinating PKA activity. It is also possible that other signaling pathways, such as TORC1, may be involved in modulating growth and metabolism phenotypes under anaerobic xylose conditions [103], though our work thus far has not investigated a role for TORC1 in these phenotypes. Finally, this study has been solely focused on the RAS branch of PKA activation, but it is also possible that the Gpa2 branch possesses an important role in growth and metabolism coupling [104]. It is clear that more studies are required to obtain a complete understanding of this mechanism.

It is evident from this and many other studies that cells have deeply intertwined the regulation of multiple processes, and disrupting one can have dramatic impacts on many others. Our results here and in previous work implicate the importance of regulatory rewiring in decoupling cellular processes. While engineering xylose metabolism pathways is essential to enable the process, anaerobic xylose fermentation is not enacted without rewiring the regulatory system to simultaneously activate Snf1 along with PKA [31,32]. Here, we propose roles for several regulators, including Opi1 and Bcy1, among downstream effectors like Azf1, Aft1/2, Mga2, and Ino2/4, in modulating growth and metabolism decoupling on anaerobic xylose. Our results strongly suggest that upstream regulatory tinkering rather than altering individual downstream effectors will be required to optimally engineer new cell functions.

## Methods

### Media and growth conditions

Cells were grown in YP media (10 g/L yeast extract, 20 g/L peptone) with 20g/L of either glucose or xylose. Aerobic cultures were grown at 30˚C with vigorous shaking. Anaerobic cultures were grown in a Coy anaerobic chamber (10% $CO_2$, 10% $H_2$, 80% $N_2$) at 30˚C with a metal stir bar for mixing. All cultures were inoculated with cells grown aerobically to saturation in YP-glucose and washed one time with the desired growth medium. Anaerobic cultures were inoculated into media incubated in the anaerobic chamber for >16 hours before inoculation. Cell density was monitored by optical density at 600 nm ($OD_{600}$) with an Eppendorf Spectrophotometer. Sugar and ethanol concentrations were measured with HPLC-RID (Refractive Index Detector) analysis [27]. Growth on solid media (Fig S5C) was performed by collecting 1 OD worth of cells from a saturated YP-glucose culture, washing cells with YP-xylose, and plating serial dilutions onto solid YP medium with 2% xylose, with or without 100 μg/mL of nourseothricin. Plates were grown in a Coy anaerobic chamber for seven days before imaging.

### Strains and cloning

*Saccharomyces cerevisiae* strains used in this study are described in Table 3. Gene knockouts were created by homologous recombination with either *KanMX* or *Hph* cassettes [105,106] and confirmed with diagnostic PCRs. The *KanMX* cassette was rescued from the *bcy1Δ* and EWY55 strains with CRISPR-Cas9 using a gRNA specific for *KanMX* and a repair template containing the flanking sequence. The *bcy1Δ* or EWY55 strain's allele of *OPI1*, *RIM8*, or *TOA1* was cloned into the pKI plasmid, carrying a nourseothricin [NAT] resistance marker, using standard cloning techniques. Plasmids were verified with Sanger sequencing, then transformed into the appropriate *bcy1Δ* or EWY55 KAN marker rescued strain using NTC selection.

### RNA-seq sample collection, RNA extraction, library preparation, and sequencing

Cells from saturated cultures of Y184, *ira2Δ*, *bcy1Δ*, and *ira2Δbcy1Δ* were used to inoculate anaerobic YPD cultures at $OD_{600}$ 0.05. Cultures grew for five hours to early/mid-log phase. 50 mL of the culture was collected, washed with YPX, then used to inoculate anaerobic YPX cultures as described above. Cold 5% phenol/95% ethanol was added to the remaining 50mL YPD cultures, which were harvested by centrifuging at 3000 RPM for 3 minutes and flash frozen in liquid nitrogen. Cell pellets were stored at -80˚C until further processing. The YPX cultures grew for 3.5 hours, when the *ira2Δ* strain resumed growth. Cold phenol/ethanol was added to the 50 mL cultures, which were harvested, flash frozen, and stored at -80˚C. Samples were collected from three independent replicates performed on different days.

Total RNA was extracted using hot phenol lysis [107] and DNA was digested with Turbo-DNase (Life Technologies, Carlsbad, CA) for 30 minutes at 37˚C. RNA was precipitated at 20˚C in 2.5 M LiCl for 30 min. rRNA was depleted with EPiCenter Ribo-Zero Magnetic Gold Kit (Yeast) RevA kit (Illumina Inc, San Diego, CA), and the remaining RNA was purified using Agencourt RNACleanXP (Beckman Coulter, Indianapolis, IN) by following the manufacturers' protocols. RNA-seq libraries were created with the Illumnia TruSeq stranded total RNA kit (Illumina) following the preparation guide (revision C), AMPure XP beads were used for PCR purification (Beckman Coulter, Indianapolis, IN), and cDNA generated with Super-Script II reverse transcriptase (Invitrogen, Carlsbad, CA) as described in the Illumina kit. Libraries were standardized to 2 μM and clusters were generated with standard Cluster kits

**Table 3. Strains used in this study.**

| Strain Name | Description | Ref |
|---|---|---|
| Y184 | CRB strain with xylose utilization genes (G418-R), *gre3*::MR *isu1*::loxP-Hyg (Hyg-R) | [32] |
| Y184 *ira2Δ* | Y184 *ira2*::MR | [31] |
| Y184 *bcy1Δ* | Y184 *bcy1*::KanMX (Hyg-R, G418-R) | [32] |
| Y184 *ira2Δbcy1Δ* | Y184 *ira2*::MR *bcy1*::KanMX (G418-R) | [35] |
| EWY55 | Y184 *bcy1*::KanMX (G418-R) *opi1*-T715G *tpk1*-G829C *toa1*-T354A *rim8*-C1591T chr. I dup. | This study |
| EWY87-1 | Y184 *bcy1*::KanMX (G418-R) chr. VIII dup. | This study |
| EWY87-3 | Y184 *bcy1*::KanMX (G418-R) *rpa43*A751G | This study |
| EWY89-1 | Y184 *bcy1*::KanMX (G418-R) *hsc82*G235C Chr. X, XVI dup. | This study |
| EWY89-2 | Chr. I, X, XVI, XIV$^{560217\text{-}625584}$ dup. | This study |
| EWY89-3 | Y184 *bcy1*::KanMX (G418-R) *hsc82*G235C Chr. I, IX, X, XVI, XIV560217-625585 | This study |
| EWY55 *opi1Δ* | EWY55 *opi1*::*KanMX* (G418-R) | This study |
| EWY55 *rim8Δ* | EWY55 *rim8*::*KanMX* (G418-R) | This study |
| EWY55 *toa1Δ* | EWY55 *toa1*::*KanMX* (G418-R) | This study |
| EWY55 empty vector | EWY55 pJH1-NatMX | This study |
| EWY55 *opi1Δ* empty vector | EWY55 *opi1*::*KanMX* (G418-R) pJH1-NatMX | This study |
| EWY55 *opi1Δ* p*OPI1*-EWY55 | EWY55 *opi1*::*KanMX* (G418-R) p*OPI1*-EWY55-NatMX | This study |
| EWY55 *opi1Δ* p*OPI1*-bcy1Δ | EWY55 *opi1*::*KanMX* (G418-R) p*OPI1*-bcy1Δ-NatMX | This study |
| EWY55 *rim8Δ* empty vector | EWY55 *rim8*::*KanMX* (G418-R) pJH1-NatMX | This study |
| EWY55 *rim8Δ* p*OPI1*-EWY55 | EWY55 *rim8*::*KanMX* (G418-R) p*RIM8*-EWY55-NatMX | This study |
| EWY55 *rim8Δ* p*OPI1*-bcy1Δ | EWY55 *rim8*::*KanMX* (G418-R) p*RIM8*-bcy1Δ-NatMX | This study |
| EWY55 *toa1Δ* empty vector | EWY55 *toa1*::*KanMX* (G418-R) pJH1-NatMX | This study |
| EWY55 *toa1Δ* p*OPI1*-EWY55 | EWY55 *toa1*::*KanMX* (G418-R) p*TOA1*-EWY55-NatMX | This study |
| EWY55 *toa1Δ* p*OPI1*-bcy1Δ | EWY55 *toa1*::*KanMX* (G418-R) p*TOA1*-bcy1Δ-NatMX | This study |

(version 3) and the Illumina Cluster station. Paired-end 50-bp reads were generated using standard SBS chemistry (version 3) on an Illumina NovaSeq 6000 sequencer.

## RNA-seq data processing and analysis

RNA-seq reads were processed with Trimmomatic version 0.3 [108] and mapped to the Y22-3 genome [58] using BWA-MEM version 0.7.17 with default settings. Read counts were calculated with HTSeq version 0.6.0 [109] using the Y22-3 gene annotations. All raw data were deposited in the NIH GEO database (GSE220465). Raw sequence counts were normalized

using trimmed mean of M-values (TMM) method [110]. log$_2$ fold changes (FC) between YP-xylose and YP-glucose samples for each strain and replicate were calculated, then hierarchical clustered using Gene Cluster 3.0 [111] and visualized with Java Treeview version 1.2.0 [112]. Differential expression was analyzed using linear modeling in edgeR version 4.0.3 [113] using pairwise and group comparisons, calling significance at < 0.05 Benjamini and Hochberg false discovery rate (FDR) [114]. Genes in Fig 2B were identified by pairwise comparisons between The Y184, *bcy1Δ*, and *ira2Δbcy1Δ* strains with the *ira2Δ* strain. Genes in Fig 2C were identified by comparing the *ira2Δ*, *bcy1Δ*, and *ira2Δbcy1Δ* strains as a group in the statistical model with the Y184 strain. Genes in Fig 3A were identified first by pairwise comparison between the *ira2Δ* and *bcy1Δ* strains, then subsequently further grouped by genes reproducibly expressed in opposing directions between the two strains (e.g. log$_2$FC > 0 in *bcy1Δ* and log$_2$FC < 0 in *ira2Δ*).

Genes differentially expressed between EWY55 and *bcy1Δ* strains grown anaerobically on xylose were identified using edgeR version 4.0.3 [113] at FDR [114] < 0.05. Genes were median centered, the log$_2$ YPX abundance of EWY55 or *ira2Δ* transcripts, relative to log$_2$ *bcy1Δ* YPX abundance were calculated, then hierarchically clustered Gene Cluster 3.0 [111] and visualized in Java Treeview version 1.2.0 [112].

Functional gene ontology (GO) term and transcriptional regulator enrichment was performed using SetRank [115]; an FDR cutoff of 0.05 was used for transcription target analysis and a Bonferroni corrected *p*-value cutoff of $10^{-4}$ was used to assess overlapping GO categories. Targets of transcription factors were downloaded from YeasTract [116] using only targets with DNA binding evidence. Upstream regulatory motifs were identified with MEME suite version 5.4.1 [117] and associated transcription factors were implicated using Tomtom [118].

## Lipidomics sample collection and preparation

Cells were grown as described previously for the RNAseq collection, flash frozen in liquid nitrogen, then stored at -80˚C. On the day of analysis, each sample was removed from -80˚C and maintained on dry ice until time of extraction. 240 µL chilled methanol was added to cell pellet samples in their native tubes over dry ice. Native tubes were transferred to ice and then vortexed. Samples were then transferred to 2 mL microcentrifuge tubes over ice. Next 800 µL of chilled methyl tert-butyl ether (MTBE) was added to native tubes followed by vortexing; these samples were also transferred to the microcentrifuge tube. Microcentrifuge tubes were then vortexed for 10 seconds. A 1/32 teaspoon (0.15 mL) of 1,180 µm glass beads (16–25 US sieve) was added to each tube along with 200 µL LC-MS grade water. Tubes were vortexed for 10 seconds. All tubes were centrifuged at 4˚C for 2 minutes at 5,000 x g to pellet cell debris. An extraction blank was prepared per sample preparation steps directly into a 2 mL microcentrifuge tube without yeast.

200 µL of the top (lipophilic) layer from each tube was aliquoted into a low volume amber borosilicate glass autosampler vial with tapered insert. For pooled YPD and pooled YPX samples, the 200 µL aliquot was performed in duplicate. Each vial was dried in a vacuum concentrator for approximately one hour. For pooled YPD and pooled YPX samples, resuspension was performed with 50 µL of a 9:1 MeOH:toluene solution on the first of two preparations ("1X") while the second preparation was resuspended in 25 µL of 9:1 MeOH:toluene ("2X"). Remaining dried samples were resuspended in 50 µL of 9:1 MeOH:toluene. Each vial was vortexed vigorously for 10 seconds to ensure resuspension of the dried contents. Samples were placed in the instrument's autosampler at 4˚C to await injection.

## Lipidomics LC-MS analysis

LC-MS/MS analysis was performed using an Acquity CSH C18 column (2.1 mm × 100 mm, 1.7 µm particle size, Waters) held at 50˚C and a Vanquish Binary Pump (400 µL/mL flow rate;

Thermo Scientific, Waltham, MA). Mobile phase A consisted of ACN:H2O (70:30, v/v) with 10 mM ammonium acetate and 0.025% acetic acid. Mobile phase B consisted of IPA:ACN (9:1, v/v) with 10 mM ammonium acetate and 0.025% acetic acid. Initially, mobile phase B was held at 2% for 2 min and increased to 30% over 3 min. In consecutive ramping steps, mobile phase B was increased to 50% over 1 minute, increased to 85% over 14 minutes, and increased to 99% over 1 minute. The gradient was held at 99% mobile phase B for 7 minutes, then decreased to 2% over 0.25 minutes. The column was equilibrated at 2% mobile phase B for 1.75 minutes before the next injection. 10 μL of each extract was injected by a Vanquish Split Sampler HT autosampler (Thermo Scientific, Waltham, MA) in a randomized order.

The LC system was coupled to a Q Exactive HF Orbitrap mass spectrometer (MS) through a heated electrospray ionization (HESI II) source (Thermo Scientific, Waltham, MA). Source conditions were as follows: HESI II and capillary temperature at 350˚C, sheath gas flow rate at 25 units, aux gas flow rate at 15 units, sweep gas flow rate at 5 units, spray voltage at |3.5 kV|, and S-lens RF at 60.0 units. The MS was operated in a polarity switching mode acquiring positive and negative full MS and MS2 spectra (Top2) within the same injection. Acquisition parameters for full MS scans in both modes were 30,000 resolution, $1 \times 106$ automatic gain control (AGC) target, 100 ms ion accumulation time (max IT), and 200 to 2000 m/z scan range. MS2 scans in both modes were then performed at 30,000 resolution, $1 \times 105$ AGC target, 50 ms max IT, 1.0 m/z isolation window, stepped normalized collision energy (NCE) at 20, 30, 40, and a 10.0 s dynamic exclusion.

## Lipidomics data analysis

The resulting LC–MS data were processed using Compound Discoverer 3.1 (Thermo Scientific, Waltham, MA) and LipiDex, an in-house-developed software suite [119]. All peaks between 0.4 min and 21.0 min retention time and between100 Da and 5000 Da MS1 precursor mass were aggregated into compound groups using a 10-ppm mass, 0.2 min retention time tolerance, a minimum peak intensity of 1x10^5, a maximum peak-width of 0.75 min, and a signal-to-noise (S/N) ratio of 3. Features were required to be 5-fold greater intensity in samples than blanks. MS/MS spectra were searched against an in-silico generated lipid spectral library. Spectral matches were required to have a dot product score greater than 500 and a reverse dot product score greater than 700. Lipid MS/MS spectra which contain acyl-chain specific fragments and contained no significant interference (<75%) from co-eluting isobaric lipids were identified at molecular species level. If individual fatty acid substituents were unresolved, then identifications were made with the sum of the fatty acid substituents. Lipid features were further filtered based on 1) presence in a minimum of two raw files, 2) a median absolute retention time deviation of 3.5, and 3) average pooled relative standard deviations of less than 30%.

Differential abundance of lipids was analyzed with linear modeling in edgeR version 4.0.3 using pairwise comparisons and a Benjamini and Hochberg [114] FDR < 0.05 to call significance. After the $log_2$FC between YP-xylose and YP-glucose samples for each strain was calculated, lipids were hierarchically cluster in Gene Cluster 3.0 [111] and visualized in Java Treeview 1.2.0 [112]. For all phosphatidylcholine moieties, a paired ANOVA with a cutoff of $p < 0.05$ was performed between *ira2Δ* and *bcy1Δ* samples. All raw and processed lipidomics data files were deposited in MassIVE database under dataset number MSV000090868.

For EWY55 lipidomics data, differential abundance of lipids was analyzed by calculating the $log_2$(fold change) ratio between YPX and YPD samples for each strain and replicate. The paired $log_2$(fold change) differences between EWY55 and *ira2Δ* or *bcy1Δ* samples were calculated, and an absolute value difference greater than 1.5 on a $log_2$ scale was called significant. Average $log_2$(fold change) of classified, significant lipid classes of interest were calculated

along with standard error, and significant differences in lipid classes were called by a paired ANOVA.

## Phosphoproteomics data

Phosphoproteomics data from Myers et al. (2019) [32] was analyzed to compare the phosphorylation of phospholipid biosynthesis enzymes. Reproducible pairwise comparisons between YP-xylose samples of strains with a $\log_2$ fold-change $>2$ were called significant.

## Inositol and choline supplementation

YP-xylose medium was prepared as described above. Myo-inositol (Sigma, Burlington, MA) was added to a final concentration of 75 μM and choline (Thermo Scientific, Waltham, MA) to a concentration of 10 mM. Anaerobic cultures were inoculated from saturated overnight cultures to an $OD_{600}$ of 0.1. Growth, xylose concentration, and ethanol concentration was monitored over 44 hours. A paired ANOVA between YP-xylose and YP-xylose-inositol-choline cultures was performed to determine significant differences between growth using a $p$ value cutoff of 0.05.

## Adaptive laboratory evolutions

*bcy1Δ* cells were inoculated in anaerobic YP-glucose medium at an $OD_{600}$ of 0.01 and grown for ~21 generations. This was used to seed a fresh anaerobic YP-glucose culture at an $OD_{600}$ of 0.01, which grew for ~7 generations. From this, a YP-2% xylose 0.1% glucose culture was seeded at an $OD_{600}$ of 0.01, then grown for ~7 generations. This process was repeated four more times before plating the culture on YP-xylose and collecting single colonies capable of growing anaerobically on xylose. Evolutions were performed in three independent cultures.

## Evolved *bcy1Δ* strain genome sequencing and analysis

Evolved *bcy1Δ* strains were grown aerobically in YP-glucose and genomic DNA was extracted using the Qiagen (Hilden, Germany) Genomic-tip 20/G kit following manufacturer's protocol. Genomic DNA was fragmented into ~200 bp fragments using a sonifier with four minutes on and one minute off while incubating on ice, repeated for a total of four cycles. DNA libraries were made using the NEBNext Ultra II DNA Library Prep Kit for Illumina protocol, using the NEBNext Multiplex Oligos for Illumina (Dual Index Primers Set 1) (New England Biolabs, Ipswich, MA). Paired-end 300 bp reads were generated on an Illumina MiSeq.

Variants in the parental *bcy1Δ* strain were identified with GATK version 4.2 (Broad Institute) and substituted into the Y22-3 reference genome as a mapping reference. Reads were mapped to the newly generated *bcy1Δ* strain genome, and variants were called using GATK version 4.2 and single nucleotide polymorphisms (SNPs) annotated with SnpEff version 5.0 and vcftools version v0.,1.12b. Only SNPs occurring in the coding region of a gene were considered for further analysis and verified with Sanger sequencing.

## Supporting information

**S1 Fig. PKA pathway mutants share similar growth and metabolism phenotypes when grown anaerobically on YPD. A-C.** Average (n = 3 biological replicates) (**A**) growth (OD600, optical density), (**B**) glucose concentration, and (**C**) ethanol concentration of *ira2Δ*, *bcy1Δ*, and *ira2Δbcy1Δ* strains grown anaerobically on rich glucose medium ($p > 0.05$, ANOVA). (TIF)

**S2 Fig. Transcriptomic profile of strains with varying xylose utilization and growth capabilities show RP transcripts are not limiting. A**. Expression of 5834 genes (rows) detected in all four strains (Y184, *ira2Δ*, *bcy1Δ*, *ira2Δbcy1Δ*), organized by hierarchical clustering of log$_2$(fold change) upon glucose-to-xylose shift, as described in Fig 2. Each column represents one of three biological replicates of the denoted strain listed above. **B**. Expression of 135 ribosomal protein genes (rows) in all four strains (Y184, *ira2Δ*, *bcy1Δ*, *ira2Δbcy1Δ*), organized by hierarchical clustering of log$_2$(fold change) upon glucose-to-xylose shift. The blue-yellow heatmap on the left represents the log$_2$(fold change) in expression upon glucose to xylose shift across biological triplicates (columns). The purple-green heatmap on the right represents the abundance of each transcript (rows) in each strain grown on glucose (G) or xylose (X), relative to the average (n = 3) abundance of transcripts measured in the Y184 YPD sample.
(TIF)

**S3 Fig. Genes expression changes specific to the *bcy1Δ* strain show changes in induction/repression and not basal mRNA abundances. A**. Expression of 654 genes whose log$_2$(fold change) upon glucose to xylose shift is different (FDR < 0.05) between the *ira2Δ* and *bcy1Δ* strains and whose change in expression is in the opposite direction (increased or decreased) across strains. (see Methods for details). The yellow-blue heatmap on the left represents the YPX/YPD log$_2$(fold change). The green-purple heatmap on the right represents transcript (rows) abundance in anaerobic glucose (G) and anaerobic xylose (X) relative to the average (n = 3) abundance of transcript in the Y184 YPD sample. **B**. ~500 base pairs upstream of the ORF for genes repressed in the *bcy1Δ* strain upon shift to xylose were analyzed for enriched motifs (bottom motif; MEME Suite), analyzed for known transcription factor binding sites (TOMTOM), and identified the Aft1/2 consensus site (top motif; see Methods for details). **C**. Expression of 77 genes from A whose promoters are physically bound by Ino2 and/or Ino4, organized by hierarchical clustering.
(TIF)

**S4 Fig. Phosphatidylcholine abundance upon shift to xylose is lower in *bcy1Δ* cells compared to *ira2Δ* cells. A**. The majority of phosphatidylcholine species (rows) identified show significantly lower log$_2$(fold change) upon the shift from glucose to xylose in the *bcy1Δ* and *ira2Δbcy1Δ* compared to the *ira2Δ* strain (*p* = 0.0015082, ANOVA).
(TIF)

**S5 Fig. Evolved *bcy1Δ* strain recapitulates the *ira2Δ* strain's phenotype but not transcriptome. A-B.** Average (n = 3 biological replicates) of **(A)** xylose concentration or **(B)** ethanol concentration for *ira2Δ*, *bcy1Δ*, and EWY55 strains anaerobically grown in rich xylose medium (*p* > 0.05, ANOVA). **C.** Representatives of multiple replicates of EWY55 or EWY55 cells lacking *OPI1* (top panels), *RIM8* (middle panels), or *TOA1* (bottom panels) and complemented with an empty vector or parental or evolved allele grown anaerobically on solid xylose (left) or glucose (right) medium with NTC selection. **D.** Average (n = 3 biological replicates) of ethanol concentration for EWY55 and EWY55 *opi1Δ* strains anaerobically grown in rich xylose medium (*p* > 0.05).
(TIF)

**S6 Fig. Directed evolution on anaerobic xylose generated multiple evolved strains with varying growth rates. A-E.** Average (n = 3 biological replicates) growth (OD600, optical density) of *ira2Δ*, *bcy1Δ*, EWY55, and **(A)** EWY87-1, **(B)** EWY87-3, **(C)** EWY89-1, **(D)** EWY89-2, or **(E)** EWY89-3 strains grown anaerobically on rich xylose medium (*p* < 10$^{-4}$, ANOVA).
(TIF)

**S1 Table. log$_2$(fold change) (n = 3) for all transcripts in dataset.** Companion table for S2A Fig.
(XLSX)

**S2 Table. log$_2$(fold change) (n = 3) for transcripts that significantly differ in fold change upon xylose shift in Y184, *bcy1Δ*, and *ira2Δbcy1Δ* cells compared to the *ira2Δ* strain.** Companion table for Fig 2B.
(XLSX)

**S3 Table. Average (n = 3) log$_2$(fold change) of cell cycle kinase and cyclin transcripts.**
(XLSX)

**S4 Table. log$_2$(fold change) (n = 3) for ribosomal protein transcripts.** Companion table for S2B Fig.
(XLSX)

**S5 Table. log$_2$(fold change) (n = 3) for transcripts that significantly differ in fold change upon xylose shift in the xylose fermenting strains (*ira2Δ*, *bcy1Δ*) compared to the Y184 strain.** Data for *ira2Δbcy1Δ* cells is also shown. Companion table for Fig 2C.
(XLSX)

**S6 Table. log$_2$(fold change) (n = 3) for transcripts that are annotated in central carbon metabolism and significantly differ in fold change upon xylose shift in *ira2Δ* and *bcy1Δ* cells compared to the Y184 strain.** Data for *ira2Δbcy1Δ* cells is also shown. Companion table for Fig 2D.
(XLSX)

**S7 Table. log$_2$(fold change) (n = 3) for transcripts that significantly differ in fold change and directionality upon xylose shift in *bcy1Δ* cells compared to *ira2Δ* cells.** Companion table for Fig 3A.
(XLSX)

**S8 Table. Functional gene ontology enrichments for clusters in Fig 3A.**
(XLSX)

**S9 Table. log$_2$(fold change) (n = 3) for Ino2/4 gene targets that significantly differ in fold change and directionality upon xylose shift in *bcy1Δ* cells compared to *ira2Δ* cells.** Companion table for S3C Fig.
(XLSX)

**S10 Table. log$_2$(fold change) (n = 3) for all lipids in dataset.**
(XLSX)

**S11 Table. log$_2$(fold change) in abundance (n = 3) for lipids that significantly differ in fold change upon xylose shift in *ira2Δ*, *bcy1Δ*, and *ira2Δbcy1Δ* cells compared to the Y184 strain.** Companion table for Fig 4A.
(XLSX)

**S12 Table. log$_2$(fold change) in abundance (n = 3) for lipids that significantly differ in fold change upon xylose shift in *ira2Δbcy1Δ* cells compared to the *ira2Δ* strain.** Companion table for Fig 4B.
(XLSX)

**S13 Table. log$_2$(fold change) in abundance (n = 3) for all phosphatidylcholine entities.** Companion table for S4 Fig.
(XLSX)

**S14 Table. Transcripts whose abundance significantly differs in EWY55 cells compared to the *bcy1Δ* strain in xylose.** Transcript abundance differences (n = 3) of EWY55 and *ira2Δ* compared to *bcy1Δ* cells.
(XLSX)

**S15 Table. Transcripts (n = 3) whose abundance significantly differs in EWY55 and *ira2Δ* cells compared to the *bcy1Δ* strain in xylose.** Companion table for Fig 5C.
(XLSX)

**S16 Table. Transcript abundance differences of phospholipid biosynthetic genes in EWY55 and *ira2Δ* cells compared to *bcy1Δ* cells on xylose.**
(XLSX)

**S17 Table. $\log_2$(fold change) in abundance (n = 3) for lipids that significantly differ in fold change upon xylose shift in EWY55 cells compared to the *bcy1Δ* strain.** Companion table for Fig 5D.
(XLSX)

## Acknowledgments

We thank Mike Place for computational help with transcriptomic data analysis, James Hose and Venera Bouriakov for help with generating RNAseq libraries, Kevin Myers for phospho-proteomic data, and members of the Gasch lab for constructive discussions.

## Author Contributions

**Conceptualization:** Ellen R. Wagner, Audrey P. Gasch.

**Data curation:** Nicole M. Nightingale, Annie Jen, Katherine A. Overmyer.

**Formal analysis:** Ellen R. Wagner, Nicole M. Nightingale, Annie Jen, Katherine A. Overmyer, Audrey P. Gasch.

**Funding acquisition:** Ellen R. Wagner, Joshua J. Coon, Audrey P. Gasch.

**Investigation:** Ellen R. Wagner, Katherine A. Overmyer, Audrey P. Gasch.

**Methodology:** Ellen R. Wagner, Nicole M. Nightingale, Annie Jen, Katherine A. Overmyer, Mick McGee, Audrey P. Gasch.

**Project administration:** Joshua J. Coon, Audrey P. Gasch.

**Resources:** Joshua J. Coon, Audrey P. Gasch.

**Software:** Joshua J. Coon, Audrey P. Gasch.

**Supervision:** Joshua J. Coon, Audrey P. Gasch.

**Validation:** Ellen R. Wagner, Nicole M. Nightingale, Annie Jen, Katherine A. Overmyer, Mick McGee.

**Visualization:** Ellen R. Wagner.

**Writing – original draft:** Ellen R. Wagner, Audrey P. Gasch.

**Writing – review & editing:** Ellen R. Wagner, Nicole M. Nightingale, Annie Jen, Katherine A. Overmyer, Joshua J. Coon, Audrey P. Gasch.

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
