## [Decision Letter · Decision Letter 0]

16 Feb 2023

Dear Dr Gasch,

Thank you very much for submitting your Research Article entitled 'PKA regulatory subunit Bcy1 couples growth, lipid metabolism, and fermentation during anaerobic xylose growth in Saccharomyces cerevisiae' to PLOS Genetics.

The manuscript was fully evaluated at the editorial level and by independent peer reviewers. The reviewers appreciated the attention to an important problem, but raised some substantial concerns about the current manuscript. Based on the reviews, we will not be able to accept this version of the manuscript, but we would be willing to review a much-revised version. We cannot, of course, promise publication at that time.

If you decide to revise the manuscript for further consideration at PLOS Genetics, please aim to resubmit within the next 60 days, unless it will take extra time to address the concerns of the reviewers, in which case we would appreciate an expected resubmission date by email to plosgenetics@plos.org.

We are sorry that we cannot be more positive about your manuscript at this stage. Please do not hesitate to contact us if you have any concerns or questions.

Yours sincerely,

Justin C. Fay

Academic Editor

PLOS Genetics

Gregory P. Copenhaver

Editor-in-Chief

PLOS Genetics

The reviews were positive, but did bring up many questions, clarifications and suggestions. While I found no key concern that must be addressed for publication, the reviewers comments are quite thoughtful and they have suggested both experiments and changes to the manuscript that would improve the quality and impact of the work.

Reviewer's Responses to Questions

**Comments to the Authors:**

Reviewer #1: Signaling pathways control a diversity of cellular responses, from changes in cell morphology that occur during differentiation, to changes in cellular metabolism, to the response to stress. One terrific model to understand cellular signaling is the budding yeast S. cerevisiae because of the ease of genetic manipulation and the long history of characterization of its main signaling pathways. One of the major nutrient-regulatory pathways in yeast is the Ras-cAMP-PKA pathway. This pathway controls the cellular response to nutrients, particular carbon and nitrogen sources, by executing a wide variety of effector responses. In the study by Wagner et al., the role of the Ras-cAMP-PKA pathway is explored from the perspective of xylose production, a compound used in biofuels. An assortment of ‘omics’ approaches and directed evolution experiments identify proteins that couple xylose-dependent growth to metabolism.

The abstract, introduction, and discussion of the manuscript are for the most part well written and suitable for a general audience. Previous literature is cited and put into the context of the current study. In the results, multiple ‘genome-wide’ experiments were performed – transcriptomics, lipidomics, and proetomics. It is clear that these experiments appear to be carefully executed, and new biologically relevant connections have been uncovered by these approaches. For example, PKA appears to control a response to iron (known), carbon (known), fatty acid (new), and phospholipid metabolism (new). The main problem is that the manuscript comes across as more of a survey, because few of the new leads were followed up on. If one molecular mechanism were pinpointed or verified among the many suggested possibilities, it would strengthen the ms and would certainly be on par with standards of PLOS Genetics. For example, in Figures 2 and 3, the authors identify many potential routes for new regulation. Instead of pursuing these hypotheses, the authors do two more global experiments. Moreover, the two main hypotheses tested by specific experiments produced results that were ambiguous. One test in Figure 4B showed the effect of inositol on growth, which very modestly stimulated growth of the bcy1 mutant, and also appeared to stimulate growth of the ira2 mutant. The other test showed that loss of opi1 impacted the growth of an evolved strain, which for technical reasons could not be recapulated. The second major problem with the paper is the presentation of the data repeatedly forces the reader into the supplement. Reorganization of the figures may address this problem. Specific concerns are noted below.

Major points

Figure 1, the ira2 bcy1 double mutant is referred to in panel D but not panels B and C (it’s in the supplement). Perhaps this data can be shown in the figure to reduce confusion? Moreover, Fig. S1 shows ira2 and bcy1 single mutants show reduced xylose but an ira1 bcy1 double does not. How are the authors concluding the double mutant is fermenting xylose?

Figure 2 is entitled ‘Few transcriptomic changes’ however, for the non-expert, it looks like many gene levels change. Our expectation might be that 65 genes and 603 genes is a lot. Where are the two genes that changed on the heat map (DSE4 and MET2)? Similarly, where is CLN2 and CDC20 on the heat map? For 63 genes, perhaps they all can be put alongside the heat map?

RAS pathway comments:

a. Lines 617-619: “One possibility is that BCY1 deletion upregulates PKA activity to a higher level than deletion of IRA2, whose activation of PKA is indirect via cAMP regulation.” PKA is also regulated by the GPCR-G-alpha protein branch (encoded by GPR1 and GPA2). Perhaps the authors could also comment on this playing a role, or why an ira2 deletion is not equal to a bcy1 deletion.

b. The Ras pathway and the TOR pathway both control ribosomal function in response to nutrient stress and maybe should be mentioned in the paper and cited: Kunkel, J., Luo, X. & Capaldi, A.P. Integrated TORC1 and PKA signaling control the temporal activation of glucose-induced gene expression in yeast. Nat Commun 10, 3558 (2019). https://doi.org/10.1038/s41467-019-11540-y

c. This paper might also be of interest to the authors, as a similar compendium of data was generated surrounding the same pathway, and the authors (somehow) managed to tie the ‘big data’ to real biology together: Multi-omics analysis of glucose-mediated signaling by a moonlighting Gβ protein Asc1/RACK1

The authors refer to “(Fig 2C, Cluster II)” Maybe add the following “(Fig 2C, Cluster II, see II at right of figure), for clarity.

Again, for Fig 2C, the authors refer to genes not shown in the figure, forcing the reader to go to the supplement to understand the figure.

Are any of the phosphor-proteomic data biologically relevant?

a. The statement “PKA has been implicated in Azf1 phosphorylation (60)”. Does PKA phosphorylate Azf1? What was the actual finding uncovered by this global study?

b. “Together, these results indicate PKA-dependent inhibition of PS synthesis in Y184 cells.” Does the phosphorylation mean anything? Does it occur in a regulated manner?

c. Does phosphorylation on S823 on Pah1 serve a function seeing as it has not been previously noted as a phosphorylation site?

The authors often refer to ‘integrated signaling systems’; however, another possibility is that there is a cascade of failures that leads to multiple problems that causes the induction of one gene set and then another. This may not be ‘integrated’ even though it is global.

In Figure 4A, the genes could be labelled to help the reader see what lipids are changing.

Figure 4C is necessary to show the whole pathway but needs work to make it clear to the reader.

a. Why are some blue and yellow squares sharp and other hazy?

b. What does the term CL in the pink square mean?

There is a problem with the statement “bcy1Δ cells experienced a very modest but statistically significant growth improvement (p = 2.4 x 10-6, ANOVA; Fig 4D), whereas the ira2Δ strain did not.” In the figure the bcy1 mutant goes from 0.59 to 0.8 (0.21 difference). The ira2 mutant similarly goes from 1.3 to 1.42 (0.12). This difference was visually obvious although not statistically significant and should be at least mentioned in the text.

Lines 427-430: “While the bcy1∆ strain’s inability to grow anaerobically on xylose cannot be fully explained by a deficiency in phospholipid precursors, the modest improvement implicates it as a contributing factor to the phenotype. Why is the effect of supplementing inositol and choline so late in the time course? If this was to explain partially why the bcy1∆ strain grows more poorly than the ira2∆ strain in xylose, then shouldn’t supplementation allow the strains to grow similarly in the early part of the time course and remain similar for longer than without supplementation?

The directed evolution experiments are interesting but the interpretations can be tricky.

a. For example, the change that was identified in Opi1resulted in an amino acid substitution (S239A), yet the comparison explored was a whole-gene deletion of Opi1. Given the subtlety of the growth defect, might it be better to plot the growth of strains instead of dilutions shown in figure 5B, as has been done for other strains in the paper?

b. The authors concede this point and refer to strain manipulation problems. Lines 488-490: “Complementation experiments were not successful, since introducing even the empty vector into this strain complemented anaerobic xylose growth on a plate for reasons that are not clear but may suggest that the cells grow differently during drug selection.” Have the authors tested an auxotrophic marker instead of an antibiotic marked plasmid to verify this hypothesis?

c. Lines 483-484: “This strain background is derived from a wild isolate that is less genetically amendable than laboratory strains” Do the authors have any other strain backgrounds they can try and verify this in?

d. Line 486: “nor did substituting the parental alleles into the evolved strain” Have the authors tried the opposite way by substituting the mutated alleles into the parental bcy1∆ strain? This seems like a better approach to test each mutation one at a time because EWY55 has multiple mutations and a chromosomal duplication.

e. The fact that aneuploidy occurs in the evolution experiments more often than not may suggest CNV could play a role instead of a SNP mutation. Have the authors looked at what genes are present on the duplicated chromosomes to see if they may help explain the phenotype? Also, have the authors ruled out whether aneuploidy affects growth in general?

f. For Fig 5B, does deleting OPI1 also decrease growth if done in the parental strains? It could be that deleting OPI1 in any strain may reduce growth and may not be specific to EWY55. If this is the case, then deleting OPI1 in EWY55 alone is not sufficient to suggest that the allele is a causal gene for the phenotype. In that case, I think the authors should generate the OPI1 allele by making the SNP change in the parental bcy1∆ strain using CRISPR-Cas9 to verify that it is playing the role they suggest it is.

g. Figure 5C is difficult to understand. To what other figure do clusters A, B, C, D refer to?

Typos and Minor Comments

1. Figure 1, the term ‘Y184’ is not clear until the reader determines that this refers to the wild-type strain. Perhaps call it wild type and refer to the specific name in the legends.

2. Fig 2B is not referenced before 2C.

3. “genetically amendable than laboratory strains” Usually the word used is amenable, although formally speaking this word choice is correct.

4. Typos in author summary: “These individual processes have been well study, but the coordination and crosstalk between the process is not well understood.” And “implicated altered regulatory mechanisms involved in lipid metabolism correlating with decouple growth and metabolism.”

5. The term “shows expression changes in the opposite direction” seems clunky.

6. The term ‘strain-specific differences’ implies that different strain backgrounds are being evaluated. However, what seems to be examined is wild type compared to mutant.

Reviewer #2: Wanger et al. performed a descriptive multiomics study deeply delving into the role of Bcy1, the PKA regulatory subunit, in yeast physiology on xylose in anaerobic culture conditions. BCY1 deletion strain cannot grow on xylose but metabolize it anaerobically due to defects in lipid homeostasis. Together with laboratory evolution, the authors additionally found that mutations in TPK1 encoding a PKA catalytic subunit and phospholipid biosynthesis regulator OPI1 recoupled the growth on and utilization of xylose of the bcy1 deletion strain, suggesting the important role of PKA-dependent regulation of lipid metabolism in the growth of the yeast strain on xylose in anaerobic conditions.

Major concerns

1.1. L145: For the comparisons between glucose and xylose cultures of these strains, the authors sampled 3 hours after the carbon source transition (glc to xyl). Did all the 4 strains exhibit similar growth/metabolites patterns on glucose before the transfer? Readers may wonder whether the differences in growth/metabolism/transcription patterns were mainly caused by the “xylose-responsive changes” or the pre-culture (on glucose) effect. For instance, one strain exhibited a significantly slower growth and/or glucose metabolism, and consequently, the strain was transferred to xyl-anaerobic culture during glucose utilization while other strains were transferred during ethanol utilization.

1.2. It would be great if the authors presented the glucose culture profiles tracking glucose consumption, ethanol production, and other metabolites, as well as the transition time point for the preculture experiment.

2.1. Also, 3 hours after the carbon source transition (probably aerobic to anaerobic culture condition transition as well, according to L653) is long enough for the ira2 deletion strain to resume its growth on xylose but might not for other strains. This can lead to some confusion about the data and authors’ arguments in the manuscript (for instance, L182 and L218).

2.2. Similar to 1.2, the authors may want to present the data tracking substrates and metabolites during the xyl-anaerobic test cultures. Especially, near the time point of the sampling for all transcription analyses. And please display error bars on the growth profiles.

3. L487: It is difficult to see the clear difference between EWY55 and OPI1 deletion strain regarding growth on solid xylose medium (Fig 5B). Liquid culture, like Fig 5A, would be a better design for comparing their growth patterns.

4. The authors also may want to present the growth data with cell biomass (dry cell weight) and CFU, which can be more suitable than OD at 600nm to argue the core messages in this study with better resolution considering the outcome of the mutants.

Minor issue

1. Introduction: It would be great if the authors specify references per each information (sentence) rather than chunky sentences with multitudinous references.

2. L201: “, respectively” seems redundant.

Reviewer #3: In this manuscript, the authors investigate an interesting observation that they had made in a previous work that different mechanisms of activating PKA (either deletion of IRA2, or deletion of BCY1) effect the ability of an engineered strain to ferment xylose and grow. Specifically the IRA2 deletion increases cAMP levels which then causes activation of PKA. This allows the engineered strain to ferment xylose and grow robustly. In contrast, deletion of BCY1, the regulatory subunit of PKA which allows the active subunits of PKA to become constitutively active regardless of cAMP levels, allows some fermentation of xylose, but has very slow growth.

This is a very interesting and important phenotype which these authors have published before and understanding how nuances in the regulation of PKA can allow metabolism and growth to be decoupled is of broad general interest. The fact that they do it in a model system (anaerobic xylose fermentation) which may have useful applications in bioproduction from renewable plant matter is also appealing. The authors produce an important and high-quality transcriptomic dataset, and a lower quality lipidomic dataset and analyse these data to make progress in understanding this mystery. Although their hypothesis that a bottleneck in lipid production is preventing growth in the BCY1 deletion strains is intriguing, I am not fully convinced by the evidence for this. They complement this systems-phenotypic approach with an investigation of strains that have overcome slow growth during laboratory evolution. Although they identify a few potential causative mutations, and find evidence that one mutation is relevant to their phenotype, I am not convinced that this mutation is they key reason that the adapted strain has regained growth, and given that the other strains they recovered have different mutations, there may be many other potential mutations that would allow the strain to retain growth.

Despite these limitations, I believe this study is worthy of publication provided that the authors moderate some of their claims and ensure that the text accurately reflects the data they present. I also found several places where the manuscript was difficult to read and confusing because of poor and possibly inaccurate word choice.

The authors first present transcriptomic data comparing anaerobic growth of strains with deletions in IRA2 and BCY1 or both genes with either xylose or glucose as a carbon source. They first compare expression in the IRA2 delete (the one strain that could grow) vs. the other three (which could not) and only find two genes differentially expressed in that strain vs the other three analysed as a group. They find that the gene expression of the IRA2 deletion strain is more similar to the non-fermenting parental strain than the fermenting BCY1 deletion strain. They do see that IRA2 deletion results in weakly repressed cyclin CLN2 and CDC20 expression while those genes are more strongly expressed in the other strains, but they suggest that these differences are not sufficient to explain the observed differences in growth.

Strikingly, the authors observe “no correlation of RP and RIBI transcript abundance or response with growth phenotypes”, as the BCY1 delete and IRA2/BCY1 double delete show very little decrease in RP and Ribi transcripts in response to xylose, while the IRA2 delete, which is growing, has a strong decrease.

The authors then compare expression in the non-fermenting strain to the fermenting strains and find many more genes that are differentially expressed. They find that several key metabolic genes are induced strongly in the parental strain but not in the deletion mutants consistent with the hypothesis that ‘the three xylose fermenters recognize xylose as a fermentable carbon, whereas Y184 [the parental strain] does not’.

The next comparison was between the BCY1 deletion strain and the IRA2 deletion strain to see how growth and metabolism were decoupled. In this comparison, the authors looked for enrichment of various functional terms, targets of transcriptional regulators and presence of predicted Transcription Factor motifs in a gene’s promoter. This analysis implicated various regulators with more or less well studied links to PKA including INO4 which is involved with phospholipid biosynthesis. This observation and the fact that many lipid biosynthesis genes were present in genes differentially expressed between BCY1 and IRA2 deletion strains led the authors to conduct lipidomics assays on the cells.

The lipidomics assays were not very reproducible between replicates, but the authors argued that there was a trend in which PC lipids were lower after the xylose shift in BCY1 deletion cells compared to IRA2 deletion cells. They argued that this combined with transcriptomic evidence and phorphorylation status indicated a bottleneck in the PC lipid biosynthesis pathway in BCY1 delete cells. They attempted to rescue this bottleneck with phospholipid precursors, but this only had a minor effect on growth of BCY1 delete cells in xylose.

The authors then conducted laboratory evolution in BCY1 Deletion strains to see if they could identify mutations that rescue growth in xylose. They found several colonies with differing point mutations, copy number variations and aneuploidies that grew better in xylose including one (EWY22) that grew even better than the IRA2 deletion strain. They found that deleting one of the genes altered in that strain caused the evolved strain to grow slightly more slowly.

They then gathered transcriptomic and lipidomic data in that evolved strain and surprisingly the transcriptome and lipidome of that strain was very different from that of the IRA2 delete strain that could also grow in xylose. They argued that the lipidome of the EWY strain overcame the hypothesized bottleneck in PC production in some way distinct from IRA2 delete cells.

The discussion has a very nice section describing how PKA induction may be different between the IRA2 deletion strain and the BCY1 deletion strain. I find myself wishing that the authors had pursued some of these hypotheses more with their experiments to get at why those two perturbations that presumably act in the same direction on the same pathway contribute to the phenotypic differences they observe in the different strains.

I believe the following points are critical problems that ought to be addressed before publishing:

1. The authors state that the IRA2/BCY1 double deletion mutant behaves phenotypically like the BCY1 double deletion mutant, but Fig S1B shows that it actually does not ferment xylose unlike both single deletion strains (Line 142). In fact, the authors state that the double deletion strain is fermenting. Is there an error in FIG S1B? If not, what is the evidence that that strain ferments xylose? I believe that this difference was not pointed out anywhere else in the text. I wrote the rest of this review as though the IRA2/BCY1 is fermenting xylose, as that is an assumption underpinning much of the interpretation of the paper, but I am not convinced that that assumption is true, at least based on the presented evidence.

2. The authors interpret their lipidomic and transcriptomic analysis (e.g. in Line 388) as generating a bottleneck in BCY1 deletion strains in the pathway leading to PC synthesis. However PDME lipids immediately upstream of PC are shown to increase actually increase in the BCY1 deletion mutants. Lower expression of OPI3 in that pathway might support their interpretation, but if so the authors should explain why that gene’s repression can lead to an increase in one of its products (PDME) and a decrease in the other (PC). I admit I am not well versed in lipid biosynthesis, but the omics evidence, combined with the fact that only a very minor increase in growth was achieved when lipids were added back to relieve the bottleneck makes it hard to believe that interpretation. Also when the authors measure the lipidome in the evolved strain which can achieve growth, they suggest that the bottleneck has been overcome (because they have increased levels of DG an TG storage lipids), however there are still decreased amounts of some of the key lipids that they implicate in the bottleneck. The authors suggest that the bottleneck is therefore overcome in some unknown way, but perhaps this evidence should cause them to question the existence or nature of the bottleneck in the first place.

In general for the lipidomic measurements (which I admit I am not very familiar with), it is disconcerting that the authors are making conclusions based on very few examples that are confidently classified (in Fig 4A, of the 18 lipids that changed, only two seem to have been confidently classified, and it seems to be a similar case for the other lipidomic analysis). It seems unwise to draw conclusions based on these few confidently classified lipids when the majority of changes are from lipids that are not confidently classified.

Finally, these lipidomic measurements seem less reproducible (i.e. than the transcriptomic measurements). The authors helpfully point this out in line 369 mentioning that one of their three BCY1 deletion replicates was not similar to the other two. In that case the authors ought to show the data from all the replicates as they did for the transcriptomic data in Figs 2 and 3.

They did show the replicates in Fig S4, but in the text they said that the lower abundance of PC lipids was ‘reproducible’ (lines 386-388) which was not very apparent to me in the figure. In one of the replicates of BCY1 the PC lipids seem to have increased according to that figure.

A minor related point on this topic: as the comparison described in the text is between the IRA2 delete and the BCY1/IRA2 double delete, the key for Fig 4C should indicate that lipids colored in green were higher in the double delete rather than higher in the single delete (similar for the pink label).

3. I am not convinced that the point mutation the authors saw in the OPI1 gene in strain EWY55 is causal for the growth phenotype. First, it would be helpful if growth of the OPI1 deletion strain were shown over time (as in Fig 5A) so one could evaluate how much of an effect that deletion has on the phenotype. Secondly, the authors point out that there is a chromosome 1 duplication. It seems that this may have an important effect on the phenotype. The authors should at least acknowledge in the text that the chromosome duplication may be important for the phenotype. Could the inability to conduct complementation experiments because introduction of the empty vector could complement low anaerobic growth on xylose on a plate under drug selection be related to this? Perhaps the drug treatment or transformation protocol could give rise to chromosome instability which could lead to restored growth in xylose.

4. One of the more striking findings of this paper is that the Ribosomal Protein genes and Ribosome Biogenesis genes are not correlated with growth in the various mutants. This is shown in Fig S1B. It seems possible that some of these genes might have altered basal expression in the deletion strains. That information would be good to show in fig S1B (similar to what is shown in fig 2C). In fact, one can see that there is a subset of genes related to translation that have reduced basal expression in the deletion strain. Also, there seems to be a group of translation related genes that have reduced expression in the two strains with BCY1 deletions relative to the IRA1 deletion strain. I would want to see these basal changes before fully believing the claim that the BCY1 deletion strains are unlikely limited by the abundance of RP and RiBi transcripts. Perhaps volcano plots in the supplemental figures showing that there is little change in basal expression in glucose between the different strains for all genes (or at least for these translation related genes) would help to allay this concern.

5. In line 220 the authors assert that “there was not a clear gene expression pattern to describe why IRA2 delete cells grow and BCY1 delete cells do not”. I think that the authors should provide a better rationale (perhaps from literature) to explain why the difference they see in cell cycle regulator expression (e.g. CLN2 and CDC20) is not sufficient to explain the difference between the IRA2 and BCY1 deletion strains growth phenotypes. They may be just a few genes, but it is possible that they are important genes. According to SGD, CDC20 is an essential gene and conditional mutants cause cell cycle arrest in M-phase. The cleanest proof would be an experiment in which expression of these regulators is inducibly decreased, or the protein is inducibly degraded in the strains of interest. This would be a difficult experiment, so perhaps the authors would prefer to qualify their assertion to acknowledge that it is possible that these regulators could be the causal difference in expression that they were looking for, but that they don’t have evidence either way.

6. One very intriguing hypothesis in the manuscript is that the xylose fermenting strains do not sense xylose as carbon starvation. This hypothesis could stand to be further explored within the data, literature or even with further experiments. One small thing that would be to display the data from Table S6 as a barplot, or even better over a map of the metabolic pathways covered (similar to fig 4C).

A minor related point is that table S6 (and possibly other tables) rely on a missing external link for the gene name.

The following points are meant to help improve the manuscript, but should not hold up publication.

M1. In general the text could use more careful editing for grammar and sentence construction. In several places it appears that the text does not accurately describe what it intends to which makes it hard to follow.

Examples:

a. Line 197-199: We specifically investigated the set of 65 genes encoding cell-cycle regulators… I understood that the 65 genes were not just regulators but were any differentially expressed gene. I think the authors meant to say ‘investigated the set of 65 genes to identify if any encoded cell-cycle regualtors..’ or something like that.

b. I don’t understand what the statement in line 153 ‘differences in xylose response drive the differences in fold change response’ it seems circular.

c. There is a statement in line 452/453 that contradicts what is shown in the plot. Supplement. The authors state that strain EWY89-1 did not differ significantly from the BCY1 deletion strain, but the plot suggests that it did. Also in fig S6E is BCH1 del really not significantly differnt different from EWY 89-3? This looks like a mistake.

d. Lines 512-515. Hard to follow and unclear if it was referring to the clusters A and D in fig 5C or not. The number of genes analysed in each barplot for fig 5C would be helpful to show.

M2. Much of the data described is displayed in heatmaps of various groups of genes clustered in different ways. It is often hard to keep track of how each subset of genes is being chosen. A text label or visual guide in the figures might make it easier to more quickly understand these figures. Also for specific statements in the text volcano plots, scatter plots or bar plots might better illustrate the points. Heatmaps are great for summarizing data, but summaries depicting variability and statistical significance would help better assess specific claims.

A few examples:

a. In line 182, the authors state that only two genes showed xylose responsive expression changes that were specific to IRA2 delete cells. Could those be highlighted or specifically analyzed?

b. Line 232 states that ‘One group of 18 genes (Fig 2C cluster II) is the only cluster in which all three xylose-metabolizing mutants showed one pattern, gene repression, whereas Y184 induced gene expression”. If there is gene repression in the IRA2del mutant in this cluster it is very faint compared to that of the other two fermenting strains. I would need to see something more quantitative (e.g. barplot) to believe this statement.

c. Line 404 states that Fig 3A shows that BCY1 delete and IRA2/BCY1 delete cells uniquely induce PA biosynthesis genes.

d. The statement in 519-520 describing transcript abundance in EW55 cells vs deletion mutants (possibly a barplot would help)

e. In fig 5D it is not clear that the DG is higher in the EW55 strain (line 525). Also the greater induction of TG lipids is not clear – in a few cases it looks just as high as in the BCY1 deletion. It is unclear how the various values for TG from the lipidomics experiments are combined to conclude that TG is greater EWY55 as shown in fig S7 and stated in the text.

f. In the conclusion targets of MGA2 are described as being altered in BCY1 delete compared to the

g. In the conclusion, line 582, the fact that INO2/4 targets are repressed is part of the argument for a bottleneck in phospholipid biosynthesis. A separate plot for just these targets might help make this more convincing.

M3. Some of the methods for analysing the data could use more detailed descriptions (e.g. in the methods section)

E.g. what does a comparison to a set of values 'as a group’ mean – are they averaged as though they were replicates before or is this done through a contrast in the edgeR package?

M4. It is hard to interpret the regulatory relationships shown in fig 3B in light of the fact that PKA should be increased in both IRA2 deletion mutants and BCY1 deletion mutants, but the expression of these targets downstream of PKA under a shift to xylose are moving in opposite directions. It would be more helpful to get a picture of how the authors propose that the different ways of increasing PKA activity cause opposing signals to be propagated to these known or suspected PKA targets. Some nice ideas are mentioned in the following paragraph relating to lipid biosynthesis.

Given the lack of enrichment of MGA2 targets (Line 300), it seems unwise to show the link in Fig 3B without stronger evidence. The caption for that figure states that it is showing ‘transcription factors whose targets or known binding sites were enriched in A’. The targets for MGA2 were not enriched, were the known binding sites enriched? If the presence of the canonical target OLE1 is sufficient, then why do a majority of the other targets not get induced? More importantly can the authors show that Mga2 is more active in xylose shift in IRA2 delete cells but less active in a BCY1 deletion strain. This claim is repeated in line 561 referring to fig 3A in the conclusion.

M5. It would be useful to include enrichment calculations for Table S8 (e.g. how many total genes are in the dataset related to the terms, how many are in the subset of genes analysed, and enrichment calculations such as a hypergeometric test statistic).

M6. It is a bit hard to interpret the transcriptional data in figure 4C. Much of this could be because the bold boxes represent significant differences with respect to IRA2 deletion, but not significance in terms of fold change for that sample.

The following statements that rely on 4C were hard to evaluate:

a. Line 336, ‘the gene encoding the PS synthase CHO1 was strongly induced in Y184 cells'

b. Line 396 that the cardiolipin biosynthetic genes were more lowly induced in ira2 deletion cells but repressed or induced in BCY1 deletion cells, especially CLD1.

M7. In table 2 a column showing the growth rate of the strain would be helpful. Also, it is not clear from the text or caption whether this is all the SNPs and rearrangements that were found, or just a selection of potentially interesting ones.

M9. PS is listed in the text in line 526 but not shown in Fig 5D.

**Have all data underlying the figures and results presented in the manuscript been provided?**

Reviewer #1: Yes

Reviewer #2: Yes

Reviewer #3: Yes

PLOS authors have the option to publish the peer review history of their article (what does this mean?). If published, this will include your full peer review and any attached files.

Reviewer #1: No

Reviewer #2: No

Reviewer #3: **Yes: **Benjamin M Heineike

---

## [Decision Letter · Decision Letter 1]

9 Jun 2023

Dear Dr Gasch,

Thank you very much for submitting your Research Article entitled 'PKA regulatory subunit Bcy1 couples growth, lipid metabolism, and fermentation during anaerobic xylose growth in Saccharomyces cerevisiae' to PLOS Genetics.

The manuscript was fully evaluated at the editorial level and by independent peer reviewers. The reviewers appreciated the attention to an important topic but identified some concerns that we ask you address in a revised manuscript.

We therefore ask you to modify the manuscript according to the review recommendations. Your revisions should address the specific points made by each reviewer.

Yours sincerely,

Justin C. Fay

Section Editor

PLOS Genetics

Gregory Copenhaver

Editor-in-Chief

PLOS Genetics

Two of the reviewers have assessed the manuscript again and appreciate the new data, analyses and other amendments to the work. Overall they are positive. However, they also brought up additional issues and clarifications. Rev 3 in addition to minor points, brought up the variable xylose metabolism of the double mutant with respect to the lipid data, and some questions about the new experiment in fig 5B. While I don’t think collecting additional data would obviously make things more clear, I also think it is reasonable to acknowledge the issues raised in order to give the reader some idea of potential limitations. For example, the double mutant variability could mean certain lipid differences were missed (not significant). Rev 1 also made some suggestions for clarify, but these are certainly discretionary.

Reviewer's Responses to Questions

**Comments to the Authors:**

Reviewer #1: Why not include the alternative GEF Sdc25 in Figure 1A?

The p-values are clear but why are there not error bars on the graphs for Figs 1B and 1C and other growth curve data?

Compared to the very nice heat maps, Figure 4C remains difficult to interpret. There are multiple squares. Some are outlined, others are not. Can the figure be clarified for the reader?

Reviewer #3: I have reviewed the authors response and appreciate the additional experiments and clarifications they have done in the manuscript in response to my comments and those of the other reviewers. I feel like the manuscript has improved and remains of general interest to the field.

I am, however worried that the lipidomics analysis (Figure 4) relies too much on a strain (ira2/bcy1 double mutant) that has a variable xylose metabolism phenotype. Also the new data in fig 5B shows the EWY55 strain growing to much higher densities than any other strains in the paper which is worrying.

I believe both these points ought to be addressed prior to publication.

After rereading the manuscript and their responses, I have the following remaining comments:

Major points:

1) I appreciate that the authors have attempted to verify whether the ira2/bcy1 double mutant ferments xylose. However it is very unsatisfying that that mutant has such a variable xylose metabolism phenotype:

These data were recollected as part of our revisions: the new data show that

the ira2bcy1 double mutant has a variable xylose metabolism phenotype (on some days

consuming xylose anaerobically as the bcy1- strain and on other days not); but there is an

invariant growth arrest under these conditions.

The fact that the phenotype is not reproducible makes me worried that there is either a problem with the strain (genome instability or if it picked up additional mutations after first being created) or else with the growth protocol. It is commendable that they redid their analysis of the transcriptomic data leaving that strain out, and I am glad that it did not affect their conclusions. Unfortunately, as they had variable data in their bcy1 deletion mutant, they still rely heavily on data from the ira2bcy1 double mutant for their lipidomics analysis. The problem is that we don’t know what state this mutant was in for the data collected on that experiment. Was it fermenting xylose or not on that day? This seems like it would effect the metabolism significantly, which I assume would effect the lipidome.

The authors also rely on the double mutant to understand gene expression patterns correlated with growth phenotypes (section starting on line 193) which is less problematic because the double mutant does show a strong growth phenotype despite the variable fermentation phenotype.

It seems like there are a few options:

- Get to the bottom of the variably xylose metabolism phenotype and be clear about what state (xylose fermenting or not) the lipidomics data was collected in.

- Conduct the lipidomics analysis using only the two replicates of the bcy1 deletion mutant. This may make it harder to get statistically significant conclusions, but it seems better than relying on data from a strain with an ambiguous and highly variable xylose fermentation phenotype.

- Ideal, but may not be possible for this paper: Repeat the lipidomics experiment with the bcy1 deletion mutant to get enough replicates upon which to base the conclusions and leave out the data from the double deletion strain, or highly caveat any analysis based on that strain.

In a related point, Figure 4 only shows two replicates of the bcy1deletion data. It is not clear to me whether the data from one of the replicates (the outlier) was thrown out in the analysis altogether (i.e. it was determined that something went wrong with the protocol) or if it was included in the analysis but not shown on the figure. I personally think if it was included in the analysis, it would be best to include it in the figure.

2) While the new experiment in fig 5B is consistent with the author’s interpretation that the OPI1 point mutation is an adaptive change that the EW55 strain acquired to resume growth in xylose despite the bcy1 mutation, I have several worries about that data and experiment:

A. EW55 grows much better in figure 3B than in Figure 3A, though they are ostensibly the same growth conditions. Indeed the scale of the OD for figure 3B seems to be higher than for any other instance of anaerobic growth in xylose. Could there be an undetected difference in the growth conditions? In particular the final OD of the EWY55, OPI1 delete strain is actually higher than the final OD of EW55 in figure 3A.

B. Could it be that in the process of deleting the OPI1 gene in the EW55 strain, that strain also lost all or part of its duplicated chromosome 1? In that case the effect could be to either the OPI1 gene, the loss of the duplicated chromosome, or a combination of the two factors.

Minor points:

Besides those two comments I have a few more minor issues with the remaining text:

1) The authors added figure S1 to make the point in line 150 that: The three strains grow indistinguishably on glucose (p> 0.05, S1A Fig) with similar glucose consumption (p > 0.05, S1B Fig) and ethanol production (p > 0.05, S1C Fig).

I am not exactly sure how the ANOVA was implemented for these growth curves, but to the eye it looks as though the IRA2 delete cell grows slightly worse in glucose, and has lower glucose consumption and ethanol production toward the latter half of the growth curve. In general the growth curves (Fig S1A seem very variable, especially given that these are the mean values of three replicates). If the ANOVA test was done for all the timepoints on the growth curve, I wonder if there might be a bias towards the earlier part of the curve when the differences were much smaller. I have a feeling that at 8 hours and with more replicates, there would be a statistical difference between these strains (at least for IRA2delete) for these phenotypes.

As the gene expression data was collected after the cells were grown on glucose for only 5 hours, when the differences do look small, I presume that this difference will not alter their conclusions. Perhaps qualifying the ‘indistinguishable’ statement with ‘indistinguishable between 4 and 6 hours’ would be more accurate.

2) The new text in Line 161 says:

There were major differences in expression comparing the strains growing on xylose,

whereas only mild expression differences were observed comparing strains grown on glucose (see Fig 2C, right panel).

Referring to Figure 2C as evidence for this statement is somewhat circular as the 292 genes chosen for that figure were selected because they had differences in expression in the glucose to xylose shift between Y184 and the other two robustly fermenting strains. I imagine that would enrich for genes that change a lot under xylose between the strains. More convincing for this point would be to look at expression differences between strains grown in glucose for all genes (e.g. differences in basal expression values between strains for all the genes with data – thus showing basal data for S2A).

3) In figure S3B two separate motifs are shown. It is not clear from the figure or the caption what those motifs are. It appears from that the short one on the top is the AFT1/2 consensus site taken from the Meme Suite Tomtom program (possibly Yamaguchi-Iwai Y et al., EMBO J, 1996 Jul 1;15(13):3377-84), and I am guessing that the long, lower motif was empirically identified using MEME on the promoters of the genes repressed in the BCY1 delete strain on a shift to Xylose.

4) Line 353: I assume the 18 lipids were identified out of the 239 confidently identified lipids rather than the overall total of 4000 – it would be helpful to clarify that in the text.

5) Line 361 states:

Instead, we analyzed previous phosphoproteomic data from our lab and discovered that Cho1 was phosphorylated to a much higher degree in the Y184 strain on serine 46 (|log2FC| > 2, Table 1)

Perhaps I am reading table 1 incorrectly but it appears that the average log2 fold change for Cho1 S46 in Y184 compared to the two deletion strains was lower than 2: 1.08 (vs ira2del) or 1.43 (vs bcy1del)?

6) Line 534: Although many of the same genes were affected, expression in the EWY55 strain was actually more dissimilar to ira2del than the parental bcy1del strain (Fig 5C)

It is not clear to me how Fig 5C shows this. I feel like a comparison between EWY55 and IRA2 delete strain is missing.

7) In response to one of my minor comments from the review:

Line 232 states that ‘One group of 18 genes (Fig 2C cluster II) is the only cluster in which all

three xylose-metabolizing mutants showed one pattern, gene repression, whereas Y184

induced gene expression”. If there is gene repression in the IRA2del mutant in this cluster it is very faint compared to that of the other two fermenting strains. I would need to see somethingmore quantitative (e.g. barplot) to believe this statement.

Author's response:

Again, we respectfully point out that these results are all based on robust statistical analysis.

While this group of genes is not a major point of the paper, that there is statistically significant enrichment of protein folding chaperones is interesting and potentially useful for some readers.

After the authors redid the analysis, cluster II now contains more genes and the statement is on Line 252:

Cluster II contained 31 genes induced in Y184 and repressed in both ira2Δ and bcy1Δ strains. This group was enriched for genes involved in protein folding (p = 9.49x10-6, hypergeometric test)

While it is clear from the heatmap that these genes are repressed in the BCY2 delete mutant, the authors seem to claim that these genes are all repressed in IRA2 delete mutant. That does not appear to be the case. Instead it looks like the cluster shows no change in IRA2, an increase in Y184 and a decrease in BCY2 in general. I could not easily find the genes identified in cluster 2 in the supplemental table so I could not verify this.

8) I suggested in my review that the following statement could be better illustrated with a bar graph:

Line 545: In general, EWY55 cells showed lower transcript abundances of phospholipid

biosynthesis genes compared to the bcy1Δ strain grown anaerobically on xylose (S16 Table), 547 making its expression even more divergent from the ira2Δ strain.

Due to the number of phospholipid biosynthesis genes, it is not feasible to show the genes as a bar plot, so we direct the reader to S7 Fig which visualizes the comparison of transcript

abundance in EWY55 cells compared to bcy1Δ cells and S16 Table containing quantitative

data.

a) I believe that S16 Table does not have transcript abundance data, but rather lipidomics data.

b) I was not suggesting a bar plot for every single phospholipid biosynthesis gene, but rather a summary – one bar for each strain summarizing transcript abundances of all phospholipid biosynthesis genes (ideally normalized first for each gene). Alternatively the authors could state the numerical levels of transcript abundance, or the percentage of phospholipid biosynthesis genes that were higher in EWY55 compared to bcy1delta in the text, and a statistical test to show that they are different. This is a minor point – just something that as a reader I cannot assess easily based on the data shown and just have to take on faith, and the information shown on a small subset of genes highlighted in S7 Fig.

**Have all data underlying the figures and results presented in the manuscript been provided?**

Reviewer #1: Yes

Reviewer #3: Yes

PLOS authors have the option to publish the peer review history of their article (what does this mean?). If published, this will include your full peer review and any attached files.

Reviewer #1: No

Reviewer #3: **Yes: **Benjamin Heineike

---

## [Editor Report · Decision Letter 2]

22 Jun 2023

Dear Dr Gasch,

We are pleased to inform you that your manuscript entitled "PKA regulatory subunit Bcy1 couples growth, lipid metabolism, and fermentation during anaerobic xylose growth in Saccharomyces cerevisiae" has been editorially accepted for publication in PLOS Genetics. Congratulations!

Yours sincerely,

Justin C. Fay

Section Editor

PLOS Genetics

Gregory Copenhaver

Editor-in-Chief

PLOS Genetics

Comments from the reviewers (if applicable):

**Data Deposition**

http://datadryad.org/submit?journalID=pgenetics&manu=PGENETICS-D-22-01474R2

**Press Queries**

---

## [Editor Report · Acceptance letter]

2 Jul 2023

PGENETICS-D-22-01474R2 

PKA regulatory subunit Bcy1 couples growth, lipid metabolism, and fermentation during anaerobic xylose growth in *Saccharomyces cerevisiae*

Dear Dr Gasch, 

We are pleased to inform you that your manuscript entitled "PKA regulatory subunit Bcy1 couples growth, lipid metabolism, and fermentation during anaerobic xylose growth in *Saccharomyces cerevisiae*" has been formally accepted for publication in PLOS Genetics! Your manuscript is now with our production department and you will be notified of the publication date in due course.

With kind regards,

Zsofia Freund

PLOS Genetics

On behalf of:
